

**Retrospective analysis of 2015-2017 winter-time PM$_{2.5}$ in China: response to emission**
**regulations and the role of meteorology**
Dan Chen[1*], Zhiquan Liu[2*], Junmei Ban[2], Pusheng Zhao[1], and Min Chen[1]
[1]Institute of Urban Meteorology, China Meteorological Administration, Beijing, 100089, China
[2]National Center for Atmospheric Research, Boulder, CO, 80301, USA

*   *Corresponding author: Dr. Zhiquan Liu (liuz@ucar.edu) and Dr. Dan Chen (dchen@ium.cn)*



**Abstract**
To better characterize the anthropogenic emission relevant aerosol species, the GSI-WRF/Chem data
assimilation system was updated from the GOCART aerosol scheme to MOSAIC-4BIN scheme. Three year
(2015-2017) winter-time (January) surface $PM_{2.5}$ observations from 1600+ sites were assimilated hourly using
the updated 3DVAR system in the assimilation experiment CONC_DA. Parallel control experiment that did
not employ DA (NO_DA) was also performed. Both experiments were verified against the surface $PM_{2.5}$
observations, MODIS 550-nm AOD and also 550-nm AOD at 9 AERONET sites. In the NO_DA experiment
using 2010_MEIC emissions, modeled $PM_{2.5}$ are severely overestimated in Sichuan Basin (SB), Central China
(CC), YRD (Yangzi River Delta), and PRD (Pearl River Delta) which indicated the emissions for 2010 are not
appropriate for 2015-2017, as strict emission control strategies were implemented in recent years. Meanwhile,
underestimations in Northeastern China (NEC) and Xin Jiang (XJ) were also observed. The assimilation
experiments significantly reduced the high biases of surface $PM_{2.5}$ in SB, CC, YRD, and PRD, and also the
low biases in NEC. However the improvement of the low biases in XJ is relatively small due to the large
differences between the observations and the model background in the DA process, likely indicating that the
emissions in the model are seriously underestimated in this region. Assimilating surface $PM_{2.5}$ also
significantly changed the column AOD and resulted in closer agreement with MODIS data and observations
at AERONET sites.
The observations and the reanalysis data from assimilation experiment were used to investigate the year-
to-year changes. As the differences of the reanalysis data (CONC_DA) among years reflect combining effects
of meteorology and emission and the differences of modeling result from control experiment (NO_DA, with
same emissions) among years reflect the separate effect of meteorology, the important roles of emission and
meteorology in driving the changes in the three years can be distinguished and analyzed quantitatively. The
analysis indicated that meteorology played different roles in 2016 and 2017: the higher pressure system, lower
temperature and higher PBLH in 2016 are favorable for pollution dispersion (compared with 2015) while the
situation is almost the opposite in 2017 (compared with 2016) that leads to the increasing $PM_{2.5}$ from 2016 to
2017 although emission control strategies were implemented in both years. There are still large uncertainties
in this approach especially the inaccurate emission input in the model brings large biases in the analysis.





## 1. Introduction

Anthropogenic $PM_{2.5}$ (fine particulate matter with aerodynamic diameters less than 2.5 µm) is known as a robust indicator of mortality and other negative health effects associated with ambient air pollution. $PM_{2.5}$ components are complicated not only from primary emissions but also from secondary formations from various precursors (e.g. $SO_2$, $NO_x$, VOCs). Regional haze with extremely high $PM_{2.5}$ concentrations (exceeding the WHO standard tenfold) has become the primary air quality concern in China, especially over the northern China (e.g. Wang *et al.* 2014a, 2014b; Han *et al.* 2015; Sun *et al.* 2015). To control the $PM_{2.5}$ pollution and improve the overall air quality, a series of strict pollution control strategies have been implemented by the government since 2010, such as *Guiding Options on Promoting the Joint Prevention and Control of Air Pollution to Improve Regional Air Quality* (The Central Government of the People's Republic of China, 2010), *Atmospheric Pollution Prevention and Control Action Plan* (The Central Government of the People's Republic of China, 2013), in which it regulated that the environmental-related equipment (Flue-gas desulfurization and Selective Catalyst Reduction, exhaust dust removal etc.) are mandatory for industries and vehicles. In addition to the long-term pollution control strategies, different emergency measures under different pollution alerts were also implemented occasionally. For example, large industrial sources (coal-burning, cement) were under limited production to reduce emission, construction sites were restricted to prevent fugitive dust pollution, traffic restrictions were implemented on even- and odd-numbered license plates etc. Those emission control strategies were even stricter and implemented more often in northern China in winter-time when the haze events occurred more frequently. These control strategies were expected to bring significant precursor (e.g. $SO_2$, $NO_x$) and $PM_{2.5}$ emission reductions.

Although with those strict emission control strategies, the ambient $PM_{2.5}$ concentrations in major cities still fluctuated in winter-time from year to year. For example, the overall January $PM_{2.5}$ concentrations in 74 cities generally decreased from 2015 to 2016, but the concentrations in January 2017 were still higher than that in 2016 (*Ambient Air Quality Monthly Report 2015-01/2016-01/2017-01*, http://www.cnemc.cn/kqzlzkbgyb2092938.jhtml). While annual emission reduction trends were expected





from 2015 to 2017, the overall increase of surface concentrations in January 2017 is kind of contradictory,
which may indicate other factors (especially meteorology) in addition to emission may play important roles.
Some studies attempted to investigate the variability of air pollution and also the effects of climate changes
on winter-time air pollution by using statistical data. Li *et al.* (2016) indicated that wintertime fog-haze days
across central and eastern China have close relation with East Asian Winter Monsoon; Zuo *et al.* (2015)
concluded that significant weakening (strengthening) Siberia high and East Asia trough are the two main key
factors for the extreme cold events and extreme warm events over china in winter while warm boost air
pollution. In addition to statistical methodology, it's necessary to distinguish the roles of emissions and
meteorology to further investigate the driving factors of the inter-annual air pollution changes.

10        Regional air quality models are important tools, either scientifically to understand the formation of hazes

or technically to make forecasts, or evaluate the effects of control strategies. For regional modeling studies,
emission inventory is an important part to reflect the emission input in the atmosphere. Generally, emission
inventory is based on the "bottom-up" methodology relying on the statistics of energy activity and emission
factors etc. However, uncertainties in energy statistics caused variations in the emission estimates (Zhao *et al.,*
2017; Hong *et al.,* 2017; Zhi *et al.*, 2017). For regional model application, the total emissions based on
statistics are then spatially-temporally distributed according to relevant factors (He, 2012). While the
occasional emission control strategies implemented in winter time caused large uncertainties in not only the
total emission estimation but also the spatially-temporally allocations, which would lead to large biases in the
model simulations.

20        In addition to the uncertainties of emission inventory, the deficiencies in chemistry also caused model

uncertainties. Recently, more and more observations revealed that the anthropogenic emission relevant aerosol
species, such as sulfate, nitrate and ammonium (denoted as SNA) are the predominant inorganic species in
$PM_{2.5}$ in China. Observations during the winter of 2013 (e.g. Wang *et al.*, 2014c) and autumn of 2014 (Yang
*et al.*, 2015) show that SNA increases rapidly during the highest haze episodes over the Northern China Plain
(NCP) and makes up approximately half of the total $PM_{2.5}$ mass. However, the WRF/Chem model failed to



reproduce the highest PM$_{2.5}$ concentrations due to missing heterogeneous/aqueous reactions with either
GOGART (Goddard Chemistry Aerosol Radiation and Transport, Chin *et al.*, 2000, 2002) or MOSAIC (Model
for Simulating Aerosol Interactions and Chemistry)-4BIN aerosol schemes. In Chen *et al.* (2016, hereafter
Chen16), we added three heterogeneous reactions (SO$_2$-to-H$_2$SO$_4$ and NO$_2$/NO$_3$-to-HNO$_3$) in the WRF/Chem
model based on the MOSAIC-4BIN aerosol scheme. The new MOSAIC-4BIN aerosol scheme significantly
improved the simulation of sulfate, nitrate, and ammonium on polluted days in terms of both concentrations
and partitioning among those species.

8       Data assimilation (DA), combining observations with numerical model output, has proved to be skillful

at improving aerosol forecasts (e.g. Collins *et al.*, 2001; Pagowski *et al.,* 2010; Liu *et al.*, 2011; Liu *et al.,*
2016; Zhang *et al.,* 2016). Liu *et al.* (2011, hereafter Liu11) implemented AOD DA within the National Centers
for Environmental Prediction (NCEP) Gridpoint Statistical Interpolation (GSI) three-dimensional variational
(3DVAR) DA system coupled to the GOCART aerosol scheme within the Weather Research and
Forecasting/Chemistry (WRF/Chem) model (Grell *et al.*, 2005). Schwartz *et al.* (2012, hereafter S12) and
Jiang *et al.* (2013, hereafter Jiang13) extended the system to assimilate surface PM$_{2.5}$ and PM$_{10}$. Verification
results demonstrated improved aerosol forecasts from the DA system in studies over East Asia and also in the
United States.
Following Liu11, S12 and Chen16, we updated the GSI-WRF/Chem system: changing from the
GOCART aerosol scheme to MOSAIC-4BIN aerosol scheme to better characterize the complex PM$_{2.5}$
pollution in China. We applied the updated system to assimilate the PM$_{2.5}$ concentrations in January 2015,
2016 and 2017, with two purposes: 1) to reproduce the PM$_{2.5}$ trends by the DA system, and 2) to investigate
the different roles of meteorology and emissions for PM$_{2.5}$ pollution in different years. In this paper, section 2
gives model description, observations and methodology, addressing the updated GSI-WRF/Chem coupled DA
system with MOSAIC-4BIN aerosol scheme. In section 3, the assimilation results on PM$_{2.5}$ concentrations in
the January of 2015, 2016 and 2017 are presented and compared with surface observations (PM$_{2.5}$ total mass
and individual species) and also MODIS 550-nm AOD for evaluation of the DA system. Different from the



previous applications emphasizing the forecast skill improvement by the DA system, we try to make full use
of the reanalysis data to investigate the driving factors of the pollutions, and also to separate the roles of
meteorology and emissions in different years by analyzing the reanalysis data and model simulations. The
results are given in section 4. Conclusions are given in section 5.
**2.  Model description, observations and methodology**

6        The WRF/Chem settings are very similar to those of Chen16, while Chen16 focused on the Sulfate-

Nitrate-Ammonia (SNA) aerosols in Northern China Plain during October 2014 and several heterogeneous
reactions were newly added to the original chemistry modules to improve the SNA simulation performance.
The DA system used here was based upon the NCEP GSI system extended by Liu11 and S12. We assimilated
surface $PM_{2.5}$ observations and the only difference is that the MOSAIC-4Bin aerosol scheme (32 species for
PM), instead of the GOCART aerosol scheme, was chosen in the WRF/Chem model. Thus the 3-D mass
mixing ratios of those MOSAIC species at each grid point comprised the analysis (or control) variables in the
GSI 3DVAR minimization process.

14       Here, only a brief summary of the WRF/Chem configurations follows before a description of the updated

GSI DA system and settings used in this work. The important differences are noted, e.g. the observation
forward operator in the GSI system.
**2.1 WRF/Chem model and emissions**

18       As in Chen16, version 3.6.1 of the WRF/Chem model was used in this study (Grell *et al.,* 2005; Fast *et*

*al.,* 2006). The physical parameterizations employed in WRF/Chem were identical to those of Chen16 and
listed in Table 1. The Carbon-Bond Mechanism version Z (CBMZ) and Model for Simulating Aerosol
Interactions and Chemistry (MOSAIC) were used as the gas-phase and aerosol chemical mechanisms,
respectively, in this study. Aerosol species in MOSAIC are defined as black carbon (BC), organic compounds
(OC), sulfate ($SO_4$), nitrate ($NO_3$), ammonium ($NH_4$), sodium (NA) and chloride (CL) and other inorganic
compounds (OIN). We used 4 size bins with aerosols diameters ranging from 0.039-0.1, 0.1-1.0, 1.0-2.5, and



2.5-10 µm. The 24 variables in the first three bins (8 species times 3 bins) consist of the $PM_{2.5}$ total. The newly
added relative humidity (RH) dependent $SO_2$-to-$H_2SO_4$ and $NO_2/NO_3$-to-$HNO_3$ heterogeneous reactions
(details in Chen16) were also applied in the simulations.

4       The model domain with a 40-km horizontal grid spacing covers most of China and the surrounding region

(Fig. 2). There are 57 vertical levels extending from the surface to 10 hPa. The simulation started from Dec.
20 of previous year and the first eleven days were treated as a spin-up period and were not used in our analyses.

7                                    **Table 1.** WRF/Chem model configurations.

| | |
|---|---|
| Aerosol scheme | MOSAIC (4 bins) (Zaveri *et al.*, 2008) |
| Photolysis scheme | Fast-J (Wild *et al.*, 2000) |
| Gas phase chemistry | CBM-Z (Zavier *et al.*, 1999) |
| Cumulus parameterization | Grell 3D scheme |
| Short-wave radiation | Goddard Space Flight Center Shortwave radiation scheme (Chou and Suarez, 1994) |
| Long-wave radiation | RRTM (Mlawer *et al.*, 1997) |
| Microphysics | Single-Moment 6-class scheme (Grell and Devenyi, 2002) |
| Land-surface model | NOAH LSM   (Chen and Dudhia, 2001) |
| Boundary layer scheme | YSU   (Hong *et al.*, 2006) |
| Meteorology initial and boundary conditions | GFS analysis and forecast every 6 hour |
| Initial condition for chemical species | 11-day spin-up |
| Boundary conditions for chemical species | averages of mid-latitude aircraft profiles (McKeen *et al.*, 2002) |
| Dust and sea salt Emissions | GOCART |

8       As in Chen16, the Multi-resolution Emission Inventory for China (MEIC) (Zhang *et al.*, 2009; Lei *et al.*,

2011; He 2012; Li *et al.*, 2014) for January 2010 is used as the emission input. The original grid spacing of
this emission inventory is $0.25^{o} \times 0.25^{o}$ and it has been processed to match the model grid spacing (40 km).
The spatial distributions of primary $PM_{2.5}$ emission are shown in Fig. 1. The MEIC-2010 emission inventory
has already been applied in other studies (e.g. Wang *et al.*, 2014a; Zheng *et al.*, 2015) for simulations over
China for recent years. They found that this inventory provides reasonable estimates of total emissions but is
subject to uncertainties in the spatial allocations of these emissions over small spatial scales. For our
simulation, uncertainties may also arise from two other aspects: the difference between the emission base year





(2010) and our simulation year (2015-2016-2017), and the monthly allocations. As the China government has
implemented strict control strategies to insure the air quality during winter seasons since 2013, significant
emission reductions including the primary PM and precursor ($SO_2$, $NO_x$) in those strictly implemented regions
compared to the year 2010 are expected for our simulation periods. Besides, the uncertainties of the emission
allocation in the winter season would be much larger compared to other seasons. For example, Zhi *et al.* (2017)
conducted village energy survey and revealed a huge amount of missing rural raw coal for winter heating in
northern China which implies an extreme underestimation of rural household coal consumptions by the China
Energy Statistical Yearbooks.
**2.2 Updated GSI 3DVAR DA system**
NCEP's GSI 3DVAR DA system was used to assimilate surface $PM_{2.5}$ observations. The GSI 3DVAR
DA system calculates a best-fit "analysis" considering the observations (hourly surface $PM_{2.5}$ concentrations
in our case) and background fields (a 1-hr short-term WRF/Chem forecast in our case) weighted by their error
characteristics. The GSI 3DVAR DA system produces an analysis in model grid space. The analysis is obtained
through the minimization of a scalar objective function J(x) given by

$$J(\mathbf{x}) = \frac{1}{2}(\mathbf{x} - \mathbf{x_b})^{\mathrm{T}}\mathbf{B}^{-1}(\mathbf{x} - \mathbf{x_b}) + \frac{1}{2}[H(\mathbf{x}) - \mathbf{y}]^{\mathrm{T}}\mathbf{R}^{-1}[H(\mathbf{x}) - \mathbf{y}], \qquad (1)$$

where $\mathbf{x_b}$ denotes the background vector (dimension m), $\mathbf{y}$ is a vector of observations (dimension p), $\mathbf{B}$ and
$\mathbf{R}$ represent the background and observation error covariance matrices of dimensions $m \times m$ and $p \times p$
respectively. The covariance matrices determine the relative contributions of the background and observation
terms to the final analysis. $H$ is the potentially nonlinear "observation operator" that interpolates the model
grid point values to observation spaces and converts model-predicted variables to observed quantities.
**2.2.1 $PM_{2.5}$ observation operator**
In our updated DA system, GSI was used to assimilate surface $PM_{2.5}$ total mass observations. While
WRF/Chem model predicts $PM_{2.5}$ total mass in the forms of different prognostic variables depending on the
chosen aerosol scheme. As we chose the MOSAIC-4Bin aerosol schemes, the analysis variables here were the





3D mass mixing ratios of the 24 MOSAIC aerosol variables at each grid point. Model simulated PM$_{2.5}$
observations $\prod m$ were computed by summing the 24 species, given as

$$\prod m = \rho_d \sum_{i=1}^{3} [BC_i + OC_i + SO_{4_i} + NO_{3_i} + NH_{4_i} + CL_i + NA_i + OIN_i]$$

3                                                                                                        ,          (2)

where $i$ denotes the Bin numbers in the MOSAIC aerosol scheme, here the first three bins consist of the PM$_{2.5}$
total; BC, OC, SO$_4$, NO$_3$, NH$_4$, NA, CL, and OIN are black carbon, organic compounds, sulfate, nitrate,
ammonium, sodium, chloride and other inorganic compounds respectively. This formula is identical to the one
used in WRF/Chem MOSAIC scheme to diagnose PM$_{2.5}$. WRF-Chem simulated aerosol mixing ratios of the
species (inside the brackets of Eq. 2) are in $\mu g\ kg^{-1}$, so dry air density $\rho_d$ is multiplied to convert the unit
to $\mu g\ m^{-3}$ for consistency with the observations.
This speciated approach to aerosol DA within a variational system was introduced by Liu11 and further
applied by S12 and Jiang13. By using individual aerosol species as control variables, no assumptions were
made regarding the contribution of each species' mass to the total aerosol mass or shapes of the vertical profiles.
**2.2.2 PM$_{2.5}$ observations and errors**
Hourly surface PM$_{2.5}$ observations for January 2015-2017 were obtained from the China National
Environmental Monitoring Center (CNEMC). There are 1600+ sites in our modeling domain. As the 1600+
monitoring sites fall into 531 model grids, the observations within the same grid are averaged ( the latitude
and longitude too) for the purpose of statistics and verification. The observation sites (Fig. 3) spanned mostly
in the northern, central and eastern China and are relatively sparse in western China.
The observation error covariance matrix **R** in equation (1) contains both measurement and
representativeness errors. Similar to S12 and Jiang13, the measurement error $\varepsilon_0$ is defined as $\varepsilon_0 = 1.0 +$
$0.0075 \times \prod_0$  , where $\prod_0$  denotes PM$_{2.5}$ observational values (unit: $\mu g\ m^{-3}$). Following S12 and Jiang13,
representativeness errors is calculated as

$$\varepsilon_r = \gamma \varepsilon_0 \sqrt{\frac{\Delta x}{L}},$$          (3)

where $\gamma$ is an adjustable parameter scaling $\varepsilon_0$ ($\gamma = 0.5$ was used), $\Delta x$ is the grid spacing (here, 40-km)



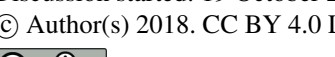

and L is the radius of influence of an observation and was set to 2-km for urban sites, respectively. The total
PM$_{2.5}$ error ($\varepsilon_{PM2.5}$) is defined as
$$\varepsilon_{PM2.5} = \sqrt{\varepsilon_0^2 + \varepsilon_r^2},$$    (4)
which constituted the diagonal elements in the **R** matrix. As those PM$_{2.5}$ data were provided in near-real time
without any data quality control. To ensure data quality before DA, PM$_{2.5}$ observational values larger than
500µg m$^{-3}$ were deemed unrealistic and not assimilated. And observations leading to innovations/deviations
(observations minus the model-simulated values determined from the first guess fields) exceeding 120 µg m$^{-3}$
were also omitted.
**2.2.3 Background error covariance**
As similar to Jiang13, the background error covariance (BEC) statistics for each analysis variable
required by the 3DVAR algorithm were computed by utilizing the "NMC method" (Parrish and Derber, 1992)
based upon the one-month WRF/Chem forecasts for the winter month of January 2015. No cross-correlation
between the different species was considered. Standard deviations and horizontal/vertical correlation length
scales of the background errors (separated for each aerosol species) were calculated using the method
described by Wu *et al.* (2002). It is important to have the phenomena-specific background error statistics to
allow for an appropriate adjustment of individual species. As a function of vertical model level, the domain-
averaged standard deviations of the background errors for 6 aerosol species (BC, OC, SO$_4$, NO$_3$, NH$_4$, OIN)
in the first three size bins are shown in Fig. 1. CL and NA are not shown here as they are relatively too small.
By using the MOSAIC aerosol schemes, the characteristic of different aerosol species in different size bins
are more appropriately described for China region in the model. As shown in Fig. 1, the standard deviations
of different aerosol species errors are different in the three size bins; the errors of NO$_3$, OIN and SO$_4$ are
relatively larger than those of other species in the three size bins; OC is also important especially in the second
(0.1-1.0 µm ) and third (1.0-2.5 µm) size bins. A larger background error of those species allowed larger
adjustment of the field, which is crucial for the aerosol analyses in this study.



**2.3 Observations for verification**
In addition to the surface $PM_{2.5}$ total mass observations for data assimilation, two types of observations
were also used for verification: (1) MODIS monthly 550-nm AOD, (2) Surface observed 550-nm AOD at
AErosol RObotic NETwork (AERONET) sites. The monthly MODIS data were downloaded from
http://ladsweb.modaps.eosdis.nasa.gov. The Terra monthly L3 dataset (daily pass time at 10:30 Local Standard
Time) was used. The data resolution is $1^{\circ} \times 1^{\circ}$. As the retrieval process in winter is much difficult than the
other seasons, there are much missing data in western and northern China. Model simulations are averaged
monthly at 03 UTC (11:00 Local Standard Time) for comparison. Actually it's also an attempt to see if the
assimilation experiment combining regional model and surface observations can generate reasonable column
AOD fields; if so, this approach can be used for a complement when the satellite data are not available in
special cases (difficult for retrieval in certain regions). The simulated 550-nm AODs at nine AERONET sites
are also compared to verify the aerosol DA performance. The locations of the nine AERONET sites are shown
as black dots in Fig. 2. The observations obtained from AERONET are interpolated to 550-nm for comparisons
(Eck *et al.*, 1999).
**2.4 Experimental design**
We conducted two sets of experiments (NO_DA and CONC_DA) for January of 2015, 2016 and 2017.
In both cases, the MEIC_2010 emission inventory was used. The NO_DA experiment initialized a new
WRF/Chem forecast every 6-hr starting 00 UTC, 20 December of previous year to spin up aerosol fields and
run through 23 UTC, 31 January. Only simulations in January were used for analysis. In the NO_DA
experiment, chemical/aerosol fields were simply carried over from cycle to cycle (similar to a continuous
aerosol forecast) while the meteorological IC/BC were updated from GFS analysis data every 6-hr to prevent
meteorology simulation drifting. For CONC_DA, GSI 3DVAR updated the MOSAIC aerosol variables every
hour starting from 00 UTC, 1 January. The background of the first cycle at 00 UTC, 1 January was from the
NO_DA experiment and the later ones were from the previous cycle's 1-hr forecast. In CONC_DA, the GFS



analysis data in 6-hr frequency were interpolated into 1-hr data and were used to update meteorological IC/BC
in each 1-hr cycle. In both the NO_DA and CONC_DA experiments, the newly added heterogeneous reactions
were all activated.
**2.5 The approach to distinguish the roles of meteorology and emission**

5        As introduced in section 1, the inter-annual air quality changes are strongly influenced by both emissions

and meteorological conditions. It's challenging to distinguish and quantify the roles of the two aspects solely
based on observation or modelling. In climate forcing studies (e.g. Xu *et al.* 2017), the role of
climate/meteorology are diagnosed by analyzing the differences between two sets of modeling simulations
(with the same emission inventory but different climate/meteorology conditions). As the emission input are
the same, the differences between the two simulations are usually attributed to the changes of
climate/meteorology fields. The approach to diagnose the role of emission is somewhat similar. Gao *et al.*
(2017) conducted WRF/Chem simulations to distinguish the roles of meteorology and emissions during the
2014 APEC week in NCP when strict emission control measures were applied. As the exact emission reduction
ratios were publicly available in BMEPB (Beijing Municipal Environmental Protection Bureau) reports for
this whole event period (before, during and after the APEC week), two simulations with different emission
scenarios (with normal and reduced emissions) but same meteorology fields were conducted. The differences
between the two simulations were attributed to the changes of emissions.

18       For our case, the same methodology can be used for meteorology aspect. As for NO_DA, the emission

input for January of the three years (2015-2017) were all from MEIC_2010 emission inventory, the only
differences among the three months' simulations were meteorological condition which was from the GFS 6-
hr analysis data. Therefore, we can assume that the differences of simulated NO_DA $PM_{2.5}$ concentrations
among the three months could be driven purely by the differences in meteorological conditions (as similar to
Xu *et al.* 2017). However, it's difficult to distinguish the role of emission by using the same approach as in
Gao *et al.* (2017). As temporary emission control measures were applied according to the pollution severity
(alarm levels) thus the emission reduction ratios were actually kept changing during the winter season and no





exact emission reduction ratios were provided for those days. The approach by simulations with different
emission scenarios is just impossible when lacking the exact emission reduction ratios. Instead, we propose
here a method by subtracting the meteorological effects from the total effects by utilizing the reanalysis data
and pure model simulations. The CONC_DA result, in which hourly surface $PM_{2.5}$ observations from 531
lumped sites were utilized, can be treated as a reanalysis dataset that reflects the actual conditions (very close
to observations). Therefore the differences of assimilated CONC_DA $PM_{2.5}$ concentrations among the three
months actually reflect combining effects of both meteorology and emissions. As the two experiments
generated gridded aerosol fields, thus we can separate the effect of emission from the total combining effects
by subtracting the NO_DA differences form CONC_DA differences. That gives us an idea how meteorology
and emission play different roles in driving the changes among the three years. Table 2 illustrates this approach
by taking 2015 and 2016 as an example. However, there might be some uncertainties in this approach which
will be discussed in detail in section 4.2.
**Table 2.** The approach to distinguish different roles of meteorology and emission by calculation from
different scenarios (take 2015 and 2016 as example).

| A. Assimilated total changes | CONC_DA_2016-CONC_DA_2015 | Reflecting the combining effects of all the driving factors from 2015 to 2016, e.g. emission, meteorology etc. |
| --- | --- | --- |
| B. Simulated changes due to meteorology differences | NO_DA_2016-NO_DA_2015 | As NO_DA_2015 and NO_DA_2016 were conducted with same emission but different meteorology, thus the differences reflect the effects from meteorological differences from 2015 to 2016 |
| C. Calculated changes due to emission differences = (A-B) | (CONC_DA_2016-CONC_DA_2015) - (NO_DA_2016-NO_DA_2015) | Mostly reflecting the effects from emission differences from 2015 to 2016 |

**3. Verification of assimilated PM$_{2.5}$**
This section presents results from the NO_DA and assimilation experiments outlined above. As $PM_{2.5}$
has significant impact on AOD, we performed verification not only against surface $PM_{2.5}$ but also against
MODIS and AERONET AOD data. Slightly different from S12 and Jiang13, our purpose is to reproduce the





spatial-temporal variations of surface PM$_{2.5}$ in the reanalysis dataset, rather than to provide the IC of aerosol
fields for improving forecasts.
**3.1 Statistics of comparison with surface PM$_{2.5}$ observations**
Figure 3 shows the observed and modeled monthly average of surface PM$_{2.5}$ for January in 2015, 2016
and 2017. Eight regions were illustrated as rectangles in the figure, including NCP (North China Plain), NEC
(Northeastern China), EGT (Energy Golden Triangle), XJ (Xinjiang), SB (Sichuan Basin), CC (Central China),
YRD (Yangzi River Delta), and PRD (Pearl River Delta). Both observation and model show that the high
values are mostly in NCP, SB and CC. In the NO_DA case, model results are over-predicted in SB, NCP and
CC for all the three months while the overestimations are more severely in SB. As the NO_DA case generally
overestimates (underestimates) surface PM$_{2.5}$ in NCP, SB and CC (XJ) in the three years, it may indicate that
the 2010 EI are not appropriate for the simulations in 2015-2017 with overestimation (underestimation)
respectively.
Compared to the NO_DA case, the assimilation experiment CONC_DA well reproduces the spatial
distribution of surface PM$_{2.5}$ for the three months, in terms of the relatively higher values in NCP, SB and CC
and also some "hot spots" in NEC, which are closer to the observations. Observations also show some "hot
spots" in XJ especially in 2016 and 2017 which are not captured by the NO_DA cases but much improved in
the CONC_DA case.
Basic statistical measures, including bias (BIAS), standard deviation (STDV), root-mean-square error
(RMSE) and correlation coefficient (CORR), are applied to evaluate the experiments. Figure 4 show the time
series of BIAS, STDV and RMSE for all the data used in the entire domain. The statistics are conducted for
each 1-hr DA cycle. After quality control, the number of PM$_{2.5}$ observations used in the DA process was
different from time to time, normally around 500-520 but with minimal of 320-450 for occasional times. The
reasons for the data filtering were from two aspects, either the PM$_{2.5}$ observational values were larger than
500 μg m$^{-3}$, or innovations/deviations (observations minus the model-simulated values determined from the
first guess fields) exceeded 120 μg m$^{-3}$, while the latter occurred more in our CONC_DA experiment. From



the time series, we can see that the bias, STDV and RMSE are greatly improved in the CONC_DA case. The
maximum biases are around 50 μg m$^{-3}$ for January 2015 and around 80 μg m$^{-3}$ for 2016 and 2017 in NO_DA,
which are reduced to around $\pm 10$ μg m$^{-3}$ in CONC_DA. The STDV and RMSE are also reduced by at least
50% for most of the times.

5        Figure 5 shows the spatial distribution of the error statistics (BIAS, RMSE and CORR) at each

observational site (with more than 2/3 valid data in the month) in January of 2015, 2016 and 2017. We start
from the comparison in 2015 and then address the differences in 2016 and 2017. In NO_DA for 2015, surface
PM$_{2.5}$ in eastern China (NCP, SB, CC, PRD and YRD) are generally overestimated by 20-60 μg m$^{-3}$, but it is
underestimated in NEC, the Energy Golden Triangle (EGT) and especially XJ. The high biases in eastern
China are greatly corrected in CONC_DA. However, the low biases in EGT and XJ still exist as most of the
observations are just filtered out in the data QC processes. That means those observations would lead to
innovations exceeding 120 μg m$^{-3}$ while such large increment probably indicates the emissions there in the
model are severely underestimated. Consistent with the BIAS changes in CONC_DA, the RMSE and CORR
in eastern China and NEC are also greatly improved with RMSE reduced by at least 50% and CORR increased
by 0.2-0.7. Without enough good observations being assimilated, the improvements in EGT and XJ are
relatively smaller. For the years of 2016 and 2017, the inhomogeneous distribution of biases in NO_DA is
very similar to 2015 (overestimated in eastern China but underestimated in NEC, EGT and XJ). However, the
high biases in CC and PRD and low biases in XJ are even larger in the latter two years. Similar to the
comparisons between NO_DA and CONC_DA for the year 2015, improvements are generally achieved except
for those sites in XJ and EGT for 2016 and 2017.
**3.2  Comparison with MODIS AOD and AERONET AOD**

22        As the improvement in surface PM$_{2.5}$ would bring changes in the optical depth, we also compare the

modeled monthly 550-nm AOD with Level-3 MODIS TERRA AOD data (Fig. 6). The MODIS AOD data are
of $1^{\circ} \times 1^{\circ}$ while model resolution is 40 km $\times$ 40 km , the different resolution between the two datasets may
bring some uncertainties in the comparison. Besides, the MODIS TERRA AOD data are missing in NEC and



western China due to the retrieval process, comparisons can only be conducted for eastern China. Spatially,
MODIS data show the high AOD values mostly in SB and CC, around 0.5-1.0. In NO_DA, the simulated
AOD reached 1.4-2.8 and even larger for SB and CC which are significantly higher than the MODIS AOD.
After assimilation, the AOD in SB and CC are significantly decreased, which are around 1.0-2.0 in the most
polluted regions. It's interesting to see that although CONC_DA did reproduce the high surface PM$_{2.5}$ in NCP
(Fig. 3), no obvious high AOD occurred there (Fig. 6c) indicating different vertical profiles of this region. The
relatively simple comparison here can't be used as evidence that the 550-nm AOD after assimilation is closer
to MODIS data, while it did show that by assimilating surface PM$_{2.5}$, the optical depth also changed greatly.

9        The simulated 550-nm AODs at nine AERONET sites (Fig. 2) are also compared with observations to

verify the aerosol DA performance. As the data are only available at several time slots with large fraction of
missing data, thus time series are not shown here. The statistics between modeled (NO_DA/CONC_DA)
experiments and the observations are listed in Table 3. At most of the sites (Beijing/Beijing-
CAMS/Xianghe/Taihu/Hong_Kong_Poly_U/Chiayi), the NO_DA and CONC_DA are all biased low, while
CONC_DA didn't correct the bias but did improve the correlations. At three sites in Hongkong and Taiwan
(Hong_Kong_Sheung/EPA-NCU/Taipei_CWB), NO_DA results are biased high and CONC_DA help to
correct the overestimation and also improve the correlation. Although there are no surface PM$_{2.5}$ observations
in the two regions, the assimilation in surrounding regions also helps due to the transport.

18                    **Table 3.** AERONET sites observed and model simulated 550-nm AOD

| Site | N Pairs of Data | MEAN | | | RMSE | | CORR | |
|---|---|---|---|---|---|---|---|---|
| | | OBS | NO_DA | CONC_DA | NO_DA | CONC_DA | NO_DA | CONC_DA |
| 1.  Beijing | 511 | 0.300 | 0.174 | 0.166 | 0.216 | 0.235 | 0.833 | 0.903 |
| 2.  Beijing-CAMS | 519 | 0.334 | 0.189 | 0.181 | 0.261 | 0.276 | 0.861 | 0.908 |
| 3.  XiangHe | 481 | 0.365 | 0.202 | 0.170 | 0.270 | 0.302 | 0.841 | 0.870 |
| 4.  Taihu | 49 | 0.278 | 0.224 | 0.122 | 0.127 | 0.187 | 0.595 | 0.833 |
| 5.  Hong_Kong_PolyU | 124 | 0.388 | 0.321 | 0.260 | 0.205 | 0.231 | 0.640 | 0.641 |
| 6.  Hong_Kong_Sheung | 39 | 0.313 | 0.642 | 0.134 | 0.486 | 0.224 | 0.520 | 0.663 |
| 7.  EPA-NCU | 58 | 0.269 | 0.470 | 0.254 | 0.390 | 0.178 | -0.001 | 0.127 |
| 8.  Taipei_CWB | 83 | 0.284 | 0.431 | 0.316 | 0.377 | 0.252 | 0.515 | 0.537 |
| 9.   Chiayi | 254 | 0.457 | 0.233 | 0.163 | 0.330 | 0.371 | 0.545 | 0.363 |



## 4. Trends in 2015-2017

Given reliable $PM_{2.5}$ reanalysis fields produced by assimilating the surface $PM_{2.5}$ (CONC_DA), changing trends among the three years can be analyzed not only on scattered observational sites but also for different regions. To distinct the roles of meteorology and emissions in driving the changes, analysis based on NO_DA and CONC_DA simulations are discussed. As assumed in section 2.5, meteorology-driven changes can be analyzed in the NO_DA simulations with different meteorology but the same emission inventory for different years; while the changes of the reanalysis data in different years are actually the combination of all the driving forces, including meteorology and emission. By analyzing the two sets of simulations, we attempted to distinguish the roles of meteorology and emissions in determining the changes.

### 4.1 Spatial distribution

The monthly-mean $PM_{2.5}$ differences for January of the three years (2015-2017) are shown in Fig. 7, in terms of surface concentrations from observational sites (Fig. 7a) and also that from assimilation experiment (Fig. 7b). Surface observations show mostly reductions from 2015 to 2016 except for a few sites in the southern parts of NCP and EGT, and also in XJ. For the changes from 2016 to 2017, surface observations show increases at almost all the sites, especially the sites in the southern part of NCP; the only exceptions are the sites along the coastline in YRD. The assimilated (CONC_DA) differences are consistent with surface observations, that the decreasing trend from 2015 to 2016 and increasing trend from 2016 to 2017 for most of the regions are reproduced. The assimilation experiment failed to reproduce the increasing trend at XJ from 2015 to 2016 as some of the highest observations were just filtered out (section 3.1) due to the large innovations in the 3Dvar process. As already shown in Fig. 3 and indicated here again, the January of 2016 is the cleanest month among the three years.

In addition to surface $PM_{2.5}$ concentrations, the spatial distribution changes of the 550-nm AOD from MODIS retrievals (Fig. 8a) and assimilation experiment (Fig. 8b) among the three years are also shown. As too much missing data in northern and western China (Fig. 6), the trends from MODIS retrievals are only available for eastern China. Yet, the MODIS 550-nm AOD changes are still overall consistent with the surface



observations, showing decreasing trend from 2015 to 2016 and increasing trend from 2016 to 2017 for the
southeastern China region. The assimilation experiment generally reproduced the trends but with some
shifting in the spatial distributions of decrease/increase regions compared with MODIS retrievals (especially
for the differences between 2017 and 2016). As the MODIS retrieval is monthly average and data filtering
were conducted day-to-day while model results were averaged for the whole month. That may lead to the
mismatch of the data period being compared.
**4.2 The roles of meteorology and emission**
Surface $PM_{2.5}$ concentrations from both observations and assimilation experiments show decreasing trend
from 2015 to 2016 but increasing trend from 2016 to 2017 for most of the regions in eastern China (Fig. 7),
which are also confirmed by the column AOD (Fig. 8). Actually, Chinese government has implemented strict
emission control strategy since 2013, especially in the northern China, and the emission reductions from year
to year are expected since 2013. Thus only justified from the emission aspect, the ambient response from
2015-2017 are just contradictory. There might be two possible assumptions: the first is the emission reduction
target was not achieved from 2016 to 2017, and the second is other factors are playing more important roles
in addition to emissions.
The NO_DA differences between different years are shown in Fig. 7c, which reflect the effect due to
meteorological condition changes (section 2.5). The effect due to emission (major factor other than
meteorology) is given by subtracting the NO_DA differences from the CONC_DA differences (Fig. 7d). We
can clearly see that the meteorology played in two different directions from 2016 to 2017. It caused decrease
in ambient concentrations for the northern China (NCP, NEC) from 2015 to 2016 but large increase for the
northern and central China (CC) from 2016 to 2017. That indicates the meteorological conditions might be
totally different from 2016 to 2017. After considering the impact from meteorology, the emission reduction is
still confirmed for the two regions from 2016 to 2017. The contributions from meteorology and emission in
the 8 regions (Fig. 3) were calculated and listed in Table 4. It shows around 13-18 $\mu g\ m^{-3}$ $PM_{2.5}$ reduction for





the month of January from 2015 to 2016 in northern China (NCP, NEC), but meteorology played the
dominating role (contributed about 12-21 µg m$^{-3}$ PM$_{2.5}$ reduction). The change from 2016 to 2017 in NCP and
NEC is totally different, meteorology caused about 12-23 µg m$^{-3}$ PM$_{2.5}$ increase and emission control measures
caused 3-13 µg m$^{-3}$ PM$_{2.5}$ decrease, that the combing effects still showed PM$_{2.5}$ increase for that region. It's
reasonable to say that the emission reductions for the northern regions from 2016 to 2017 are indeed obtained.
However, the meteorology played important role which offset the emission reduction and lead to the increase
of surface concentrations in 2017. The same approach is applied on the column AOD as shown in Fig. 8.
Consistent with surface concentrations, meteorology caused decrease/increase for northern China for the
period 2015-2016/2016-2017 respectively. The different roles of meteorology and emissions for different
regions are confirmed.
**Table 4.** Modeled PM$_{2.5}$ ambient concentration changes for 2016-2015, 2017-2016 and 2017-2015 in 8
regions, and the contributions of meteorology (MET) and emission (EMIS) calculated according to Table 2.
Unit: µg m$^{-3}$.

|  | 2016-2015 | | | 2017-2016 | | | 2017-2015 | | |
|---|---|---|---|---|---|---|---|---|---|
|  | Total | MET | EMIS | Total | MET | EMIS | Total | MET | EMIS |
| NCP | -13.38 | -12.52 | -0.86 | +9.86 | +23.16 | -13.31 | -3.53 | +10.65 | -14.17 |
| NEC | -18.06 | -21.23 | +3.17 | +9.60 | +12.61 | -3.02 | -8.46 | -8.62 | +0.16 |
| ETR | -1.90 | -3.97 | +2.07 | +7.20 | +12.94 | -5.74 | +5.30 | +8.97 | -3.67 |
| XJ | -3.29 | +0.07 | -3.35 | +5.82 | +0.28 | +5.55 | +2.54 | +0.34 | +2.19 |
| SB | -22.77 | +8.72 | -31.49 | +9.85 | +4.02 | +5.83 | -12.92 | +12.74 | -25.66 |
| CC | -15.22 | +14.12 | -29.34 | +5.13 | +20.49 | -15.35 | -10.09 | +34.61 | -44.69 |
| YRD | -9.03 | -3.03 | -5.99 | -11.65 | -2.93 | -8.73 | -20.68 | -5.96 | -14.72 |
| PRD | -24.07 | +13.02 | -37.09 | +13.20 | -6.12 | +19.32 | -10.87 | +6.90 | -17.78 |

It is worth noting that there are uncertainties in the simulation/assimilation processes. Firstly, emission
inventories are obviously not accurate in the NO_DA simulations which may bring uncertainty in the analysis.
For example, the emission in SB, CC and PRD are generally overestimated (Fig. 3), which means the ambient
concentration changes might be artificially amplified in considering the meteorology impacts (Fig. 7c and Fig.
8c). Secondly, the meteorological IC/BC conditions in NO_DA simulations, which were from GFS analysis
data every 6-hr, have also uncertainties. The biases in meteorological conditions might lead to uncertainties in





the PM$_{2.5}$ analysis. Thirdly, the accuracy of the CONC_DA assimilation experiment also affects the analysis.
For example, the assimilation did reproduce some of the "hot spots" in XJ (Fig. 3c) but can't reproduce the
increasing trends from 2015 to 2016 (Fig. 7b) as some of the highest concentrations in 2016 were not well
simulated (Fig. 3c). Finally, the contribution of aerosol-meteorology feedback was not considered in our
calculation. As pointed out by Gao *et al.* (2017), reduced aerosol feedbacks due to emission reductions account
for about 10.9% of the total decreases in PM$_{2.5}$ concentrations in urban Beijing in their APEC study. In our
current approach, this effect is combined in the emission aspects in the subtracting process.
**4.3 Meteorology changes in 2016 and 2017**

9       It's interesting to see that meteorology played different roles in the three years. Here we compared some

meteorology parameters to explain the meteorology impacts. Differences of monthly mean boundary layer
height (PBLH), surface pressure (PSFC), 2-meter temperature (T2), 2-meter relative humidity (RH2) and 10-
meter wind speed in different years are given in Fig. 9. It shows that the changes of PSFC and T2 for the
period 2015-2016 and 2016-2017 are totally different for the whole region. Compared to 2015, the pressure
system is stronger, temperature is lower, and wind speed is larger in most regions in 2016 which are favorable
for pollution dispersion. While there are some unfavorable conditions including lower PBLH and higher RH
(thus more reactions) in the northern and southern China which may offset the impacts of high pressure system
and low temperature. So the combining impacts of those meteorological parameters caused ambient
concentration decrease in northern China and increase in southern China from 2015 to 2016 as shown in Fig.
7 and Fig. 8. For the changes from 2016 to 2017, meteorological changes are totally different with weaker
pressure system, higher temperature, smaller wind speed, and lower PBLH in most regions, which caused the
pollution accumulation. As suggested by recent studies, climate change has important impacts on extreme
haze events in northern China based on historical statistical approach or by using climate models. Those
studies (e.g. Li *et al.*, 2015, Zuo *et al.*, 2015) revealed that wintertime fog-haze days across central and eastern
China have close relation with East Asian Winter Monsoon; significant weakening (strengthening) Siberia
high and East Asia trough are the two main key factors for the extreme cold events and extreme warm events





over china in winter; while warm boosts air pollution. Consistent with our study, Zhao *et al.* (2018) pointed

out that stronger Siberian High period in January 2016 produced a significant decrease in PM$_{2.5}$ concentrations

than that during the weaker ones in other years. Those studies emphasized climate change factors, the impacts

of emission changes are still difficult to evaluate. Our study used the DA technique combining regional models

and surface observations, aiming to separate the factors of emission and meteorology, thus to further

investigate the year-to-year changes for the regional scale.

## 5. Conclusions

To analyze the complex PM$_{2.5}$ pollution in China, the GSI-WRF/Chem aerosol data assimilation system

was updated from the GOCART aerosol scheme to MOSAIC-4BIN scheme, which is more appropriate to

characterize the anthropogenic emission relevant aerosol species. Three-year (2015-2017) winter-time

(January) surface PM$_{2.5}$ observations from 1600+ sites were assimilated hourly using the updated 3DVAR

system in the assimilation experiment CONC_DA. Parallel control experiment that did not employ DA

(NO_DA) was also performed.

Both the control and the assimilation experiments were verified against the surface PM$_{2.5}$ observations,

MODIS and AERONET 550-nm AOD. In the NO_DA experiment that 2010_MEIC emission inventory was

used, modeled PM$_{2.5}$ were severely overestimated in Sichuan Basin (SB), Central China (CC), YRD (Yangzi

River Delta), and PRD (Pearl River Delta) which indicated the emissions for 2010 are not appropriate for

2015-2017, as strict emission control strategies were implemented in recent years. Meanwhile,

underestimations in Northeastern China (NEC) and Xin Jiang (XJ) were also observed.

The assimilation experiment significantly reduced the high biases of surface PM$_{2.5}$ in SB, CC, YRD, and

PRD, and also low biases in NEC. However, the improvement of the low biases in XJ is relatively small as

some of the observations were filtered out in the DA system due to the large innovations which are treated as

"unrealistic"; those large innovations also indicate that the emissions in the model are seriously underestimated

in this region. Assimilating surface PM$_{2.5}$ also significantly changes the column AOD; comparisons with





MODIS 550-nm AOD showed that the control experiment without DA are too high in eastern China and that
of assimilation experiment are more close to MODIS data.
Both observation and assimilation experiment showed decreasing ambient concentration from 2015 to
2016 but increasing from 2016 to 2017 for most of the regions. To distinct the important roles driving the
changes, the reanalysis data from assimilation experiment and modeling result from control experiment were
analyzed. It shows around 13-18 $\mu g\ m^{-3}$ $PM_{2.5}$ reduction for the month of January from 2015 to 2016 in northern
China (NCP, NEC), but meteorology played the dominating role (contributed about 12-21 $\mu g\ m^{-3}$ $PM_{2.5}$
reduction). The change from 2016 to 2017 in NCP and NEC is totally different, meteorology caused about 12-
23 $\mu g\ m^{-3}$ $PM_{2.5}$ increase and emission control measures caused 3-13 $\mu g\ m^{-3}$ $PM_{2.5}$ decrease, that the combing
effects still showed $PM_{2.5}$ increase for that region. The analysis approved that meteorology played different
roles in 2016 and 2017: the higher pressure system, lower temperature and higher PBLH in 2016 are favorable
for pollution dispersion (compared with 2015); the situation is almost the opposite in 2017 (compared with
2016) that leads to the increasing $PM_{2.5}$ from 2016 to 2017, although emission control strategy were
implemented in both years. After considering the impacts from meteorology, the analysis showed that the
emission reductions were indeed obtained from 2015 to 2016 and 2017, especially in NCP for the year 2017
(although surface concentrations were increasing that year).
While there are still large uncertainties in this approach, as the inaccurate emission input, uncertainties
in the meteorological IC/BC and assimilation process, and also the imperfection of aerosol-meteorology
feedbacks in the model simulation bring large biases in the analysis. The most straightforward way is to
directly estimate the emissions by data assimilation, which will be the topic in a separate study.
**Acknowledgement**
This work was supported by the National Key R&D Program on Monitoring, Early Warning and
Prevention of Major Natural Disasters under grant (2017YFC1501406), the National Natural Science
Foundation of China (Grant No. 41807312) and Basic R&D special fund for central scientific research
institutes (IUMKYSZHJ201701). NCAR is sponsored by US National Science Foundation.



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



**Tables and Figures**
**Table 1.** WRF/Chem model configurations.
**Table 2.** The approach to distinguish different roles of meteorology and emission by calculation from different
scenarios (take 2015 and 2016 as example).
**Table 3.** AERONET sites observed and model simulated 550-nm AOD.
**Table 4.** Modeled $PM_{2.5}$ ambient concentration changes for 2016-2015, 2017-2016 and 2017-2015 in 8 regions
and the contributions of meteorology (MET) and emission (EMIS) calculated according to Table 2. Unit: µg
$m^{-3}$.
**Figure 1.** Domain-averaged standard deviations of background errors (µg $kg^{-1}$) as a function of height for
each aerosol variables in three bins: (a) Bin-01 0.039-0.1 µm, (b) Bin-02 0.1-1.0 µm, (c) Bin-03 1.0-2.5 µm.
**Figure 2.** Spatial distribution of $PM_{2.5}$ emissions (unit: µg $m^{-2}$ $s^{-1}$) used in this study. Black dots with numbers
indicate 9 AErosol RObotic NETwork (AERONET) sites used for aerosol optical depth verification: 1-Beijing
(39.98ºN, 116.38ºE), 2-Beijing-CAMS (39.93ºN, 116.32ºE), 3-XiangHe (39.75ºN, 116.96ºE), 4-Taihu
(31.42ºN, 120.22ºE), 5-Hong_Kong_PolyU (22.30ºN, 114.18ºE), 6-Hong_kong_Sheung (22.48ºN,
114.117ºE), 7-EPA-NCU (24.97ºN, 121.19ºE), 8-Taipei_CWB (25.03ºN, 121.50ºE), 9-Chiayi (23.50ºN,
120.50ºE).
**Figure 3.** Observed and modeled monthly average of $PM_{2.5}$ concentrations (unit: µg $m^{-3}$) for January in 2015
(Left), 2016 (middle) and 2017 (right). Regions defined in red rectangles are: a-NCP (North China Plain), b-
NEC (Northeastern China), c- EGT (Energy Golden Triangle), d-XJ (Xinjiang), e-SB (Sichuan Basin), f-CC
(Central China), g-YRD (Yangzi River Delta), h-PRD (Pearl River Delta).
**Figure 4.** The time series of statistics between model simulations and observations. Red lines- CONC_DA
minus observation, blue lines –NO_DA minus observation. Statistics include number of data pairs, MEAN-
mean bias, STDV- standard deviation, RMS-root mean square error. Left-2015, middle-2016, right-2017.
(Unit are µg $m^{-3}$ for MEAN, STDV and RMS).
**Figure 5.** The spatial distribution of statistics between model simulations and observations for January, (a)
2015, (b) 2016, (c) 2017. Top: NO_DA v.s. observation, bottom: CONC_DA v.s. observation. BIAS-model
minus observation, RMSE-root mean square error, CORR-correlation coefficient. (Unit is µg $m^{-3}$ for BIAS
and RMSE).
**Figure 6.** Observed and modeled monthly average of 550-nm AOD for January in 2015 (Left), 2016 (middle)
and 2017 (right). Observation (a) is from MODIS Terra monthly L3 dataset (daily path time at 10:30 Local
Standard Time). Model simulations from (b) NO_DA and (c) CONC_DA are monthly averages at 03 UTC
(11:00 Local Standard Time). (d) The difference of CONC_DA minus NO_DA.
**Figure 7.** Observed and modeled $PM_{2.5}$ ambient concentration changes for 2016-2015 (left), 2017-2016
(middle) and 2017-2015 (right). (a) Observations, (b) Assimilated total changes, (c) Modeled changed due to
meteorology conditions, (d) Calculated changes due to emission. (Unit: µg $m^{-3}$)
**Figure 8.** Similar to Figure 7 but for observed and modeled 550-nm AOD changes.
**Figure 9.** Modeled meteorological changes for 2016-2015 (left), 2017-2016 (middle) and 2017-2015 (right).
(a) PBLH, (b) PSFC, (c) T2, (d) RH2 and (e) 10-m wind speed.





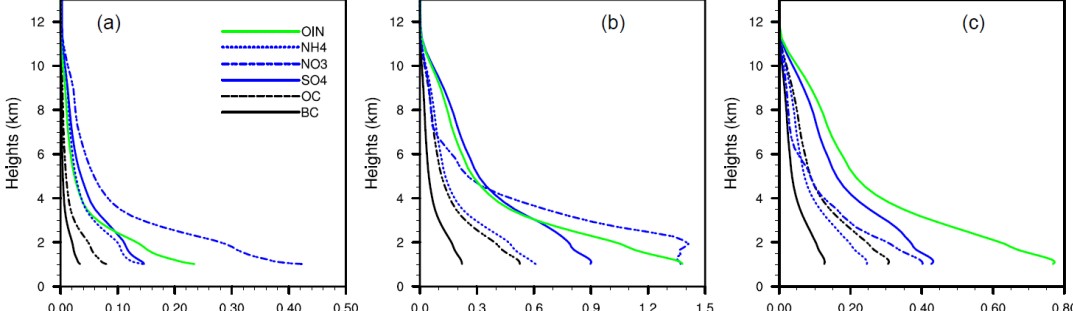

**Figure 1.** Domain-averaged standard deviations of background errors (µg kg$^{-1}$) as a function of height for each aerosol variables in three bins: (a) Bin-01 0.039-0.1 µm, (b) Bin-02 0.1-1.0 µm, (c) Bin-03 1.0-2.5 µm.





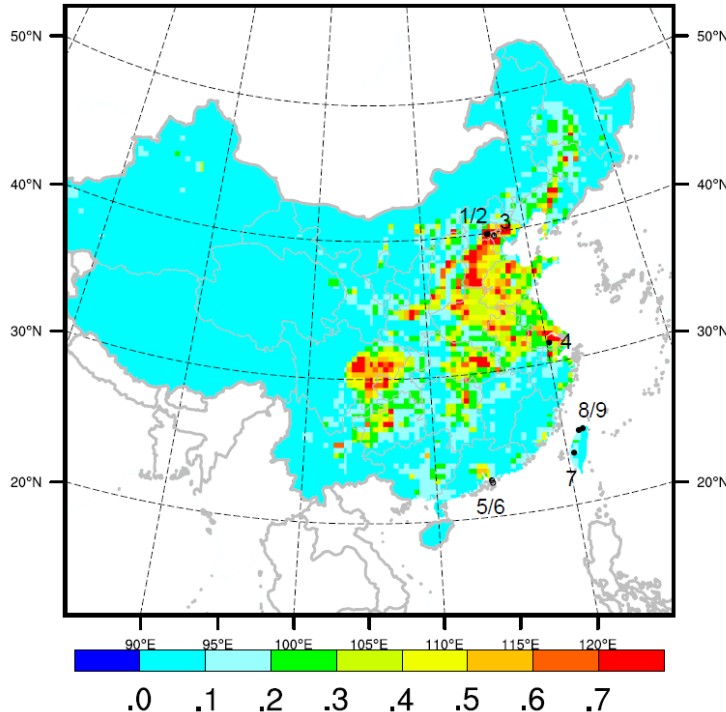

**Figure 2.** Spatial distribution of PM$_{2.5}$ emissions (unit: µg m$^{-2}$ s$^{-1}$) used in this study. Black dots with numbers indicate 9 AErosol RObotic NETwork (AERONET) sites used for aerosol optical depth verification: 1-Beijing (39.98$^{\circ}$ N, 116.38$^{\circ}$ E), 2-Beijing-CAMS (39.93$^{\circ}$ N, 116.32$^{\circ}$ E), 3-XiangHe (39.75$^{\circ}$ N, 116.96$^{\circ}$ E), 4-Taihu (31.42$^{\circ}$ N, 120.22$^{\circ}$ E), 5-Hong_Kong_PolyU (22.30$^{\circ}$ N, 114.18$^{\circ}$ E), 6-Hong_kong_Sheung (22.48$^{\circ}$ N, 114.117 $^{\circ}$ E), 7-EPA-NCU (24.97$^{\circ}$ N, 121.19$^{\circ}$ E), 8-Taipei_CWB (25.03$^{\circ}$ N, 121.50$^{\circ}$ E), 9-Chiayi (23.50$^{\circ}$ N, 120.50$^{\circ}$ E).






**Figure 3.** Observed and modeled monthly average of PM$_{2.5}$ concentrations (unit: μg m$^{-3}$) for January in 2015 (Left), 2016 (middle) and 2017 (right). Regions defined in red rectangles are: a-NCP (North China Plain), b-NEC (Northeastern China), c- EGT (Energy Golden Triangle), d-XJ (Xinjiang), e-SB (Sichuan Basin), f-CC (Central China), g-YRD (Yangzi River Delta), h-PRD (Pearl River Delta).

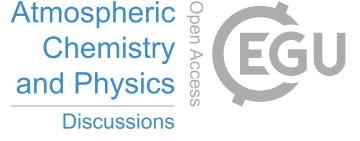

**Figure 4.** The time series of statistics between model simulations and observations. Red lines- CONC_DA minus observation, blue lines –NO_DA minus observation. Statistics include number of data pairs, MEAN-mean bias, STDV- standard deviation, RMSE-root mean square error. Left-2015, middle-2016, right-2017. (Unit are µg m$^{-3}$ for MEAN, STDV and RMS).



**(a). 2015 - NO_DA (top) and CONC_DA (bottom)**

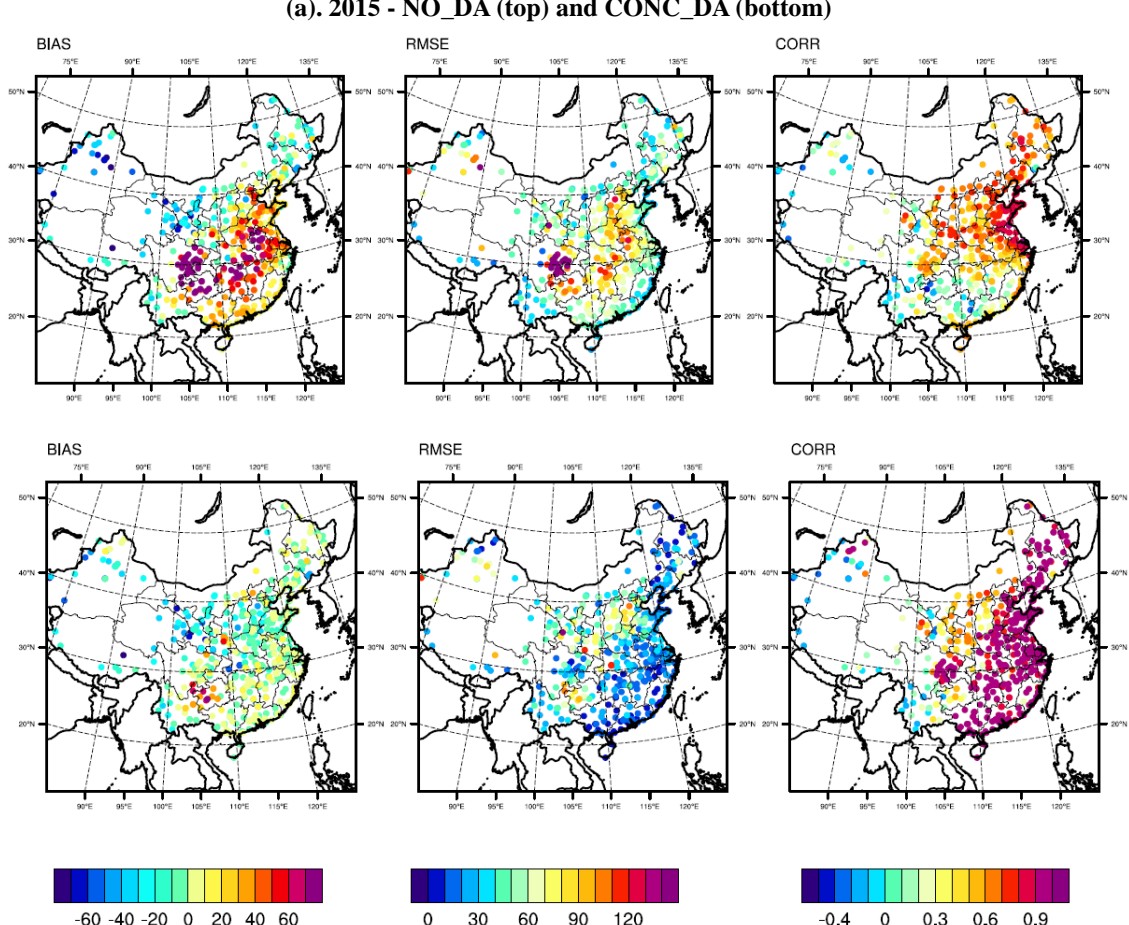

**Figure 5a.** The spatial distribution of statistics between model simulations and observations for January 2015. Top: NO_DA v.s. observation, bottom: CONC_DA v.s. observation. BIAS-model minus observation, RMSE-root mean square error, CORR-correlation coefficient. (Unit is µg m$^{-3}$ for BIAS and RMSE).



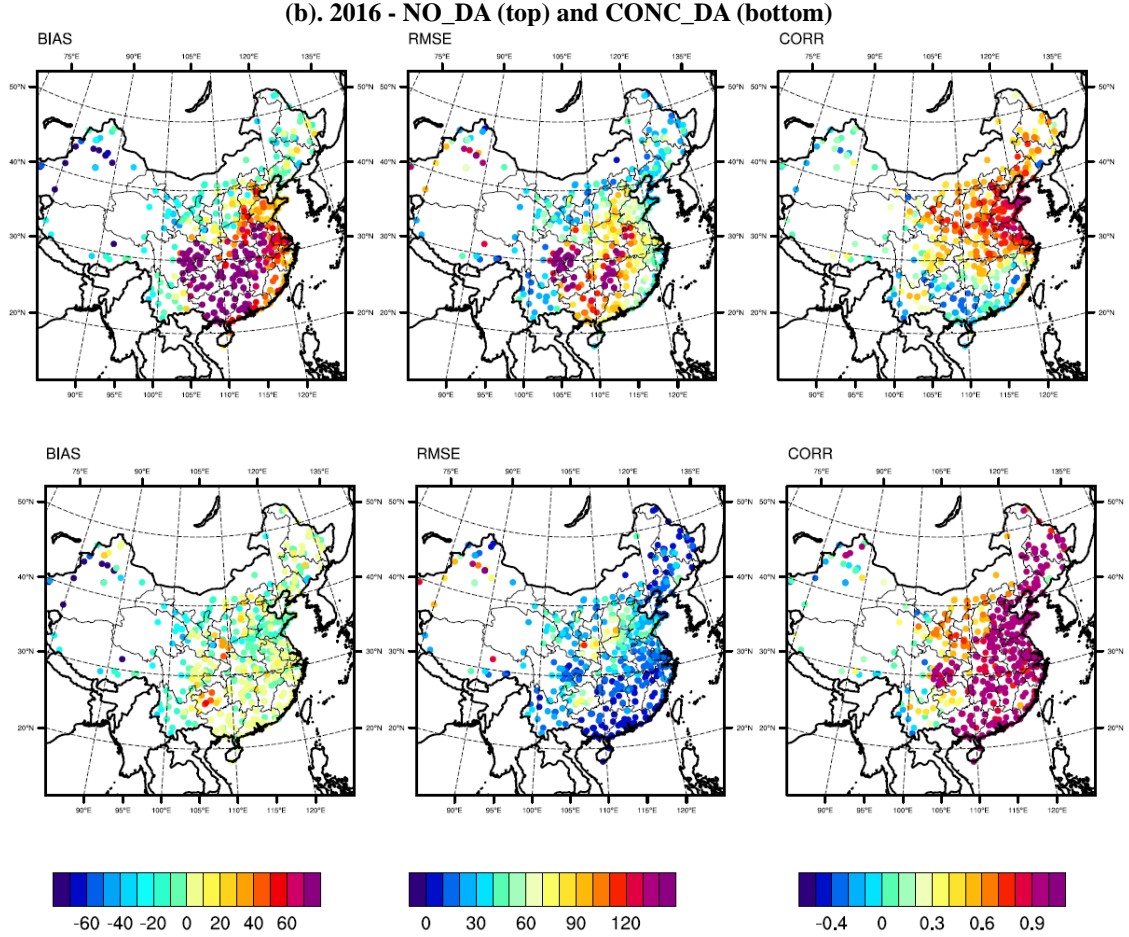

**Figure 5b.** Continue. Same as Figure 5a but for 2016.



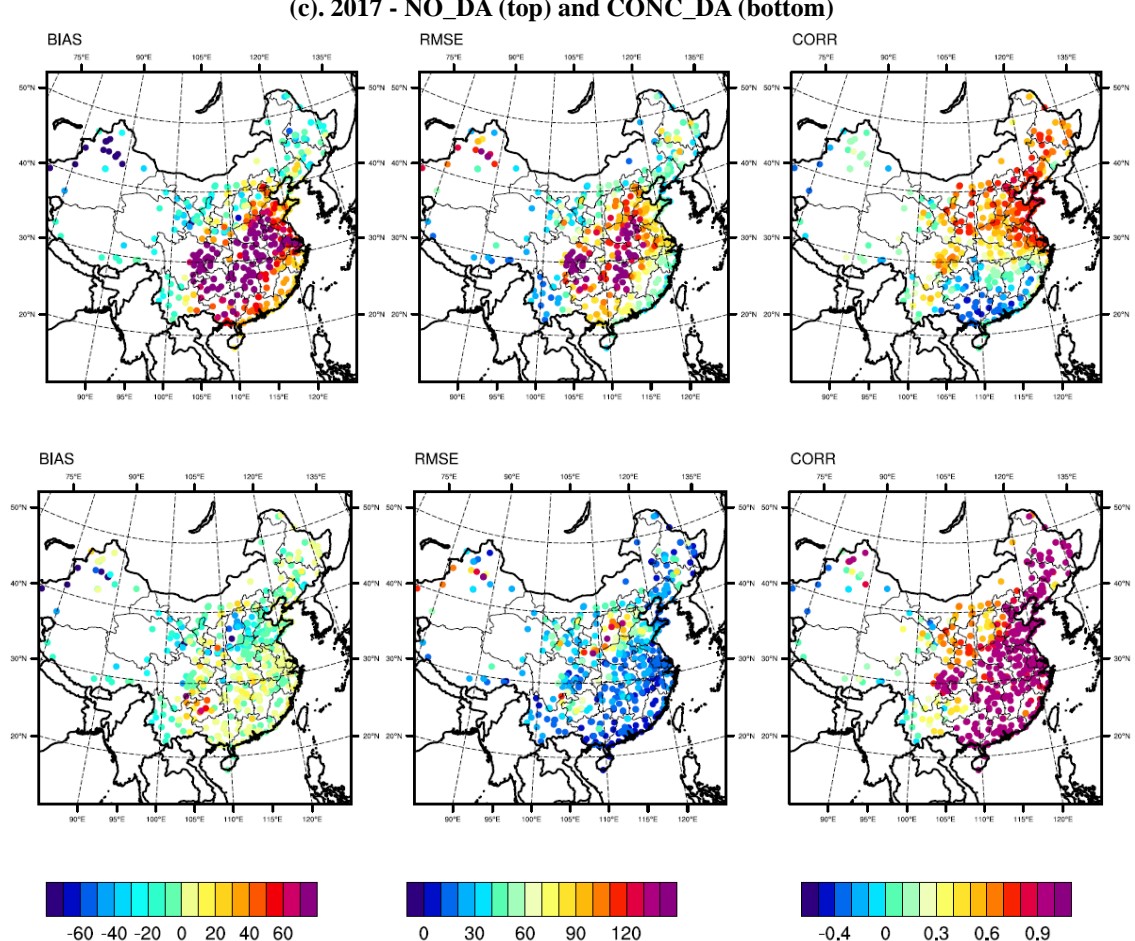

**Figure 5c.** Continue. Same as Figure 5a but for 2017.



**Figure 6.** Observed and modeled monthly average of 550-nm AOD for January in 2015 (Left), 2016 (middle) and 2017 (right). Observation (a) is from MODIS Terra monthly L3 dataset (daily path time at 10:30 Local Standard Time). Model simulations from (b) NO_DA and (c) CONC_DA are monthly averages at 03 UTC (11:00 Local Standard Time). (d) The difference of CONC_DA minus NO_DA.





**2016-2015**  **2017-2016**  **2017-2015**

**(a) Observations**

**(b) Assimilated total changes**

**(c) Modeled changes due to meteorological conditions**

**(d) Calculated changes due to emission**

**Figure 7.** Observed and modeled PM$_{2.5}$ ambient concentration changes for 2016-2015 (left), 2017-2016 (middle) and 2017-2015 (right). (Unit: µg m$^{-3}$).



**Figure 8.** Similar to Figure 7 but for observed and modeled 550-nm AOD changes.





**Figure 9.** Modeled meteorological changes for 2016-2015 (left), 2017-2016 (middle) and 2017-2015 (right) for (a) PBLH, (b) PSFC, (c) T2, (d) RH2 and (e) 10-m wind speed.