# Peer review of "Retrospective analysis of 2015-2017 winter-time PM$_{2.5}$ in China: response to emission regulations and the role of meteorology"

_Atmospheric Chemistry and Physics, 2018_

## Referee Comment (RC1) · Anonymous Referee #1 · 24 Dec 2018

This manuscript presents an interesting study that utilizes the GSI-WRF/Chem 3D variational data assimilation system to better simulate the surface PM2.5 concentrations in China for the January months 2015-2017. It shows that WRF-Chem PM2.5 simulations with assimilation of surface measurements significantly reduced the model biases and better captured the inter-annual variability of surface PM2.5 levels in January 2015-2017. The model improvements are independently evaluated with MODIS and AERONET aerosol optical depth (AOD) measurements. Comparisons of model PM2.5 simulations with and without data assimilation indicate the effectiveness of the emission control measures, as well as the unfavorable meteorological conditions in January 2017 that led to PM2.5 increases relative to January 2016.

[Figure]

Overall, I think this is a nice study that illustrates the strength of data assimilation method to constrain PM2.5 changes, and further diagnoses contributions from both emissions and meteorological conditions. The method of this study is solid, and the language is generally appropriate. I recommend publish after the following comments being addressed.

**Specific Comments:**
(1) Page 7, Line 11:
"The spatial distributions of primary PM2.5 emission are shown in Fig. 1". Here Fig 1 should be Fig 2. Does primary PM2.5 correspond to BC, OC, and OIN in the WRF-Chem model? Since PM2.5 is also produced secondary in the air, it should be useful to show its precursor emissions, such as NOx or SO2.

(2) Page 10, Line 5-8 about the data quality filter:
The study states that PM2.5 observational values larger than 500 $\mu$g m$-3$ were deemed unrealistic and observations leading to deviations exceeding 120 $\mu$g m$-3$ were also omitted. It is not clear to me how these thresholds would impact the results and the conclusions of this study. What are the fractions of data that were omitted by the filters? In winter, some cases can meet the thresholds and can be realistic. So what would happen if a looser filter was applied. Please add some discussions.

(3) Page 14, Section 3.1:
It shall be valuable to add a table in this section, similar to current Table 4, but summarizing the mean observed vs. simulated PM2.5 concentrations over the 8 regions defined in Figure 3. The readers can then have a more quantitative picture on how effective the data assimilation system is.

(4) Page 15, Section 3.2:
The use of MODIS AOD data was only for support of the AOD decreases over the
Sichuan Basin and Central China after data assimilation. This seems to be insufficient. How about the inter-annual variability of MODIS AOD over January 2015-2017? Are they consistent with surface PM2.5 measurements? Please clarify.

(5) Page 18, Line 23:
In the statement "meteorological conditions might be totally different from 2016 to 2017", "totally" is a very strong word, however, it is not clear how different 2017 meteorological conditions are different from normal wintertime conditions with Siberian Highs. I have the same comment for Page 20, Line 13, Line 19, the word "totally" is not helpful. I suggest use more quantitative statements, for example, higher temperature by how much?

(6) Page 20, Line 15:
What does "higher RH (thus more reactions)" mean? How higher RH lead to more chemical reactions? Please clarify.

(7) Captions of Figure 7 and Figure 9: Please indicate here that the comparisons are for the January month.

(8) Some technical corrections:
Page 2, Line 8 - "modeled PM2.5 are" should be "modeled PM2.5 concentrations are"
Page13, Line 7 - "reflect combining effects" should be "reflect combined effects"
Page 14, Line 11 - "the 2010 EI" should be "the 2010 emissions"
Page 19, Line 17 - "the emission in ..." should be "the emissions in ..."
* * *

---

## Referee Comment (RC2) · Anonymous Referee #2 · 9 Jan 2019

The manuscript by Chen et al. evaluates PM2.5 in January in 2015 - 2017 in China. Data assimilation is used to construct a reanalysis that well matches the available observations of surface-level PM2.5 concentrations. The difference between this and a non-forced simulation is cleverly used to parse out the separate roles of emissions from meteorology in variability of PM2.5 concentrations, which are significant across these 3 years (although they authors refer to these as trends – which I think is better described as inter annual variability). This is important and valuable work, because it helps identify reasons why emissions control strategies may or may not have an immediately visible impact. As such the topic and scope are suitable for ACP. The manuscript includes some additional analysis of AOD, but it is somewhat secondhand, not as well

supported in terms of model accuracy, and doesn't particularly add to the focus of the paper, which I suggest remain on surface-level PM2.5 concentrations. This would provide space for the authors to provide more details on other aspects of their modeling and assimilation methods, which in several places are too abbreviated or presented without sufficient background or justification. The major scientific weakness in this work is likely the use of out-dated emissions (from 2010), given the rate at which species like SO2 and NOx are known to be changing since 2010 in China. Below I provide more detailed comments on these and other aspects to address prior to publication in ACP.

Comments: The grammar needs work throughout. This might mean adding another co-author or hiring an editing service. The number of corrections needed (nearly every sentence) are far too extensive for me to detail here.

The abstract would be improved by considering some of the following suggestions:

- lead with a broader, introductory statement

- avoid jargon when possible

- try to provide a mix of qualitative and quantitative results. Presently, only qualitative results are reported

- end with a statement about the bigger impacts of this work

- reduce the overall length – currently it goes too much into details of the methods, without quantitatively summarizing the most important conclusions.

4.22: I believe that this statement regarding the inorganic PM2.5 in this region has been known for some time.

7.1-3: There is considerable debate in the literature regarding the reactions that may be leading to high concentrations in haze events. The ones included here are based on assumptions of pH that may not be correct. Other recent works have e.g. suggested HCHO may play a role. In short, I recommend the authors review the relative literature

on this topic (which should be easy to find, several very high-profile papers). Even if they decide to stick with their current mechanism (which I think is acceptable, given their is yet to be scientific consensus on this issue), discussion is warranted, both in the introduction and consideration of sources of errors towards the end of the paper.

Figure 2: It's not clear what this shows. What are "PM2.5 emissions"? Since many of the species contributing to PM2.5 are gas-phase precursors, which don't necessarily completely transform into aerosol, I'm quite puzzled. Perhaps it is a plot of a subset of PM2.5 precursor emission species, but that isn't made clear.

8.1 and other locations (e.g., 14.11): The emissions used in this work are quite outdated. It is documented in several studies of emissions in China that SO2 emissions have been decreasing since around 2009, and NOx emissions since around 2011 or 2012. These previous studies need to be cited, and considered in the present work. It is certainly a significant source of error worth considering.

9.2: Not sure why capital pi notation is used here, as that means product, where the definition is in terms of a sum.

9.10-12: I don't understand how this works – can it be explained further? The way in which measurements of total PM2.5 are used to adjust concentrations of specific species should be clarified, even though it comes from an earlier work, if only briefly.

9.20: What is the origin of this assumption regarding error? The relative error component of 0.75% seems very small. Other parameters such as gamma and L, the maximum concentration threshold (500) or analysis increment (120) are similarly introduced without much explanation. I recognize these values have been used before, but still more explanation here would be appreciated.

10.12: Omission of cross-correlation between species like ammonium and nitrate seems critical – how does this impact the results?

Section 2.5: The method used for estimating the impact of meteorology separately from

emissions seems sound; I'm not sure the extended explanation (by way of comparison to radiative forcing calculations, etc.) is needed and suggest simply removing the first paragraph of Section 2.5 and jumping directly into the statement of how this study was carried out.

Fig 4: The model performance seems good after the assimilation. One small question though – it seems like the residual bias in the CONC_DA case is most often negative. Is there a reason for this? I would have expected that, given the initial simulation is biased high, the analysis would not necessarily be unbiased, but would similarly have a small slight high bias (owing to the constraint term in the cost function).

Fig 5: Figs 5b and 5c could be omitted or moved to supporting information.

15.24: Why not use level 2 data?

16.7: I didn't follow nor agree with the logic behind this statement. Why not include statistics of the AOD comparison in a table, or directly on the figures themselves (there is room in the white space). Also, it seems that DA only serve to decrease the AOD, not increase it, even in areas.

16.14: Let's be honest – the DA didn't "didn't correct the bias" – it made the bias even worse, for 6 out of the 9 sites. This won't go un-noticed by the readers, so it might was well be presented fairly.

Overall, the AOD analysis was on the weaker side, the connections to the policy and haze questions not as clear, and the model performance not as good (especially for AERONET). I would suggest the authors consider dropping AOD related content entirely, unless some more satisfactory explanations can be included.

Section 4: Evaluation of concentrations in January alone of 3 consecutive years is not sufficient enough time range for a "trends" analysis. We do get some sense of interannual variability though, which is interesting. It is just mislabeled. For example, every place that says something like "decreasing trend from 2015 to 2016" should

say "decrease from 2015 to 2016", as the long-term trend hasn't been determined. If the authors really do wish to study trends, they should have considered years such as 2005, 2010 and 2015. Or if they can't go back that far, owing to data availability, 2012, 2014, and 2017. That would start to be close to enough years to make a trend analysis. My overall suggestion would actually be to remove section 4.1 entirely, at least the second paragraph on AOD. I also wonder if any of these years happened to be impacted by PM2.5 transport more than others, e.g. from fires?

18.13: I don't understand what is meant by the sentence beginning "Thus only…"

Fig 8: Suggest removing this figure; it is barely discussed, and doesn't add much. The analysis of PM2.5 surface concentrations is sufficient and also more convincing, since the model performance is better.

20.1: I think this is an important point – if the authors are using this approach to separate emissions impacts from meteorology in the observed dataset, then it is critical the relative changes in the total assimilation experiment (row b) match the observations.

21.14: Change "verified" to "evaluated", here and throughout, since in the strict sense of the word, the model has certainly not been verified.

22.2: I note the authors stop short of including anything regarding the AERONET evaluation in the conclusions – an indication that this part of the manuscript could be removed without impact.

---

## Author Comment (AC1) · 19 Feb 2019

**COMMENTS TO THE AUTHOR(S)**

Retrospective analysis of 2015-2017 winter-time PM2.5 in China: response to emission regulations and the role of meteorology

Manuscript ID:   acp-2018-890

Authors: Chen, et al.

**Reviewer 1**

This manuscript presents an interesting study that utilizes the GSI-WRF/Chem 3D-variational data assimilation system to better simulate the surface PM2.5 concentrations in China for the January months 2015-2017. It shows that WRF-Chem PM2.5 simulations with assimilation of surface measurements significantly reduced the model biases and better captured the inter-annual variability of surface PM2.5 levels in January 2015-2017. The model improvements are independently evaluated with MODIS and AERONET aerosol optical depth (AOD) measurements. Comparisons of model PM2.5 simulations with and without data assimilation indicate the effectiveness of the emission control measures, as well as the unfavorable meteorological conditions in January 2017 that led to PM2.5 increases relative to January 2016.

Overall, I think this is a nice study that illustrates the strength of data assimilation method to constrain PM2.5 changes, and further diagnoses contributions from both emissions and meteorological conditions. The method of this study is solid, and the language is generally appropriate. I recommend publish after the following comments being addressed.

**Response:**

We really appreciate the reviewer's thoughtful comments. It helps to improve our manuscript by addressing these issues. We have made serval changes accordingly.

1. Rerun the assimilation experiment with looser filter criteria, in which the interannual changes were better captured.

2. Added quantitative results in tables, including the statistics of control and assimilation experiments, and also the statistics of interannual differences of meteorology conditions.

Please see our itemized responses below. Revised manuscript is after the response letter.

Specific Comments:

"The spatial distributions of primary PM2.5 emission are shown in Fig. 1". Here Fig 1 should be Fig 2. Does primary PM2.5 correspond to BC, OC, and OIN in the WRF-Chem model? Since PM2.5 is also produced secondary in the air, it should be useful to show its precursor emissions, such as NOx or SO2.

**Response:**

Thanks for pointing out the typo! The figure number in the text has been corrected.

Yes, the primary $PM_{2.5}$ corresponds to the total of BC, OC, sulfate, nitrate and other PM emissions. The emission spatial distribution of $SO_2$, $NO_x$ and $NH_3$ have been added in Figure 2.

The study states that $PM_{2.5}$ observational values larger than 500 µg m$^{-3}$ were deemed unrealistic and observations leading to deviations exceeding 120 µg m$^{-3}$ were also omitted. It is not clear to me how these thresholds would impact the results and the conclusions of this study. What are the fractions of data that were omitted by the filters? In winter, some cases can meet the thresholds and can be realistic. So what would happen if a looser filter was applied. Please add some discussions.

**Response:**

Thanks for the suggestion! The criteria for filter process are from two aspects, including the stability of DA optimization step and the computing efficiency. The original criteria were mostly set for operational runs. For research purpose, we have made tests of different filter process and found that looser filter can really improve the assimilation results. Here in the revised manuscript, only $PM_{2.5}$ observations larger than 1000 µg m$^{-3}$ (the maximum display limit of the monitoring system) were deemed unrealistic in the filter process and observations leading to deviations exceeding 500 µg m$^{-3}$ were omitted. Besides, in the original assimilation experiment observational sites located in grids with elevation greater than 500 meter (Above Sea-Level) were not used; to better utilize those data, we chose to interpolate them to the lowest model level for assimilation. The data used in the two assimilation experiments increased from 3580876 (62.4%) to 5309200 (92.6%) by setting looser filters. It should be corrected that the number of data (top panel) in Figure 4 are actually the data available for comparisons, not the data used in the assimilation cycle.

Below show the observed, original and updated assimilations of monthly average of $PM_{2.5}$ concentrations (unit: µg m$^{-3}$) for January in 2015 (Left), 2016 (middle) and 2017 (right). The most prominent improvements are shown for the hotspots in Xinjiang (region d) and Fenwei Plain (added as region e).

[Figure]

**Figure S1.** Observed and modeled monthly average of PM$_{2.5}$ concentrations (Unit: µg m$^{-3}$) for January in 2015 (Left), 2016 (middle) and 2017 (right). (a) Observation, (b) Original CONC_DA, (c) New CONC_DA with looser filter. Regions defined in red rectangles are: a-NCP (North China Plain), b-NEC (Northeastern China), c- EGT (Energy Golden Triangle), d-XJ (Xinjiang), e-Fenwei Plain (FWP), f-SB (Sichuan Basin), g-CC (Central China), h-YRD (Yangtze River Delta), i-PRD (Pearl River Delta).

In the updated assimilation experiment, the interannual changes were also captured as shown below. Those improvements make our analysis more solid, especially for the Xinjiang region and Fenwei Plain. The updated figures and discussions are highlighted in blue in the revised manuscript.

[Figure]

**Figure S2.** Observed and modeled PM$_{2.5}$ ambient concentration changes for 2016-2015 (left), 2017-2016 (middle) and 2017-2015 (right). (a) Observation, (b) Original CONC_DA, (c) New CONC_DA with looser filter. (Unit: μg m$^{-3}$)

(3) Page 14, Section 3.1:

It shall be valuable to add a table in this section, similar to current Table 4, but summarizing the mean observed vs. simulated PM2.5 concentrations over the 8 regions defined in Figure 3. The readers can then have a more quantitative picture on how effective the data assimilation system is.

**Response:**

Thanks for the suggestion. Yes, we have added the statistics in Table 3.

**Table 3.** Statistics of the observed and model-simulated surface PM$_{2.5}$ for January 2015, 2016 and 2017 in 9 regions (units are μg m$^{-3}$ for BIAS and RMSE).

| Statistics | Sites | Pairs of data | BIAS | | RMSE | | CORR | |
|---|---|---|---|---|---|---|---|---|
| | | | NO_DA | CONC_DA | NO_DA | CONC_DA | NO_DA | CONC_DA |
| 2015 | | | | | | | | |
| NCP | 67 | 46699 | 19.38 | 2.08 | 68.09 | 24.26 | 0.72 | 0.96 |
| NEC | 30 | 20910 | -11.94 | -1.04 | 49.47 | 21.11 | 0.59 | 0.93 |
| EGT | 28 | 19516 | -40.43 | 5.28 | 60.62 | 19.45 | 0.37 | 0.90 |
| XJ | 19 | 13243 | -53.76 | 4.16 | 71.69 | 19.74 | 0.40 | 0.94 |
| FWP | 27 | 18819 | 4.05 | 1.75 | 56.71 | 23.05 | 0.63 | 0.93 |
| SB | 48 | 33456 | 98.02 | 0.61 | 125.76 | 20.76 | 0.55 | 0.94 |
| CC | 49 | 34153 | 46.94 | -0.38 | 81.31 | 21.18 | 0.46 | 0.93 |
| YRD | 34 | 23698 | 32.22 | -0.43 | 59.90 | 15.14 | 0.73 | 0.96 |
| PRD | 20 | 13940 | 19.36 | -0.03 | 47.81 | 9.10 | 0.24 | 0.95 |
| 2016 | | | | | | | | |
| NCP | 67 | 46699 | 20.90 | 1.41 | 57.77 | 20.74 | 0.78 | 0.96 |
| NEC | 30 | 20910 | -11.05 | 0.04 | 40.91 | 16.08 | 0.57 | 0.94 |
| EGT | 28 | 19516 | -22.55 | 0.69 | 39.63 | 13.75 | 0.42 | 0.90 |
| XJ | 19 | 13243 | -72.92 | 0.25 | 98.19 | 27.16 | 0.51 | 0.96 |
| FWP | 27 | 18819 | -3.51 | 1.51 | 62.04 | 26.01 | 0.76 | 0.94 |
| SB | 48 | 33456 | 134.63 | 2.77 | 165.38 | 15.49 | 0.51 | 0.92 |
| CC | 49 | 34153 | 86.28 | 1.89 | 109.09 | 18.76 | 0.46 | 0.92 |
| YRD | 34 | 23698 | 46.13 | 1.03 | 62.11 | 13.40 | 0.73 | 0.95 |
| PRD | 20 | 13940 | 59.79 | 2.05 | 74.76 | 6.51 | 0.04 | 0.91 |
| 2017 | | | | | | | | |
| NCP | 67 | 46699 | 25.75 | 2.35 | 82.31 | 28.91 | 0.74 | 0.95 |
| NEC | 30 | 20910 | -11.38 | 0.01 | 53.38 | 21.35 | 0.64 | 0.94 |
| EGT | 28 | 19516 | -26.88 | 1.40 | 48.83 | 16.96 | 0.41 | 0.90 |
| XJ | 19 | 13243 | -95.92 | 3.82 | 125.09 | 35.65 | 0.51 | 0.96 |
| FWP | 27 | 18819 | -6.78 | -1.02 | 89.26 | 31.69 | 0.65 | 0.94 |
| SB | 48 | 33456 | 122.82 | 2.33 | 149.08 | 20.08 | 0.56 | 0.93 |
| CC | 49 | 34153 | 101.22 | 3.49 | 132.97 | 19.50 | 0.23 | 0.92 |
| YRD | 34 | 23698 | 59.31 | 2.40 | 78.02 | 12.32 | 0.63 | 0.93 |
| PRD | 20 | 13940 | 35.01 | 0.04 | 61.84 | 9.55 | -0.16 | 0.94 |

(4) Page 15, Section 3.2:

The use of MODIS AOD data was only for support of the AOD decreases over the Sichuan Basin and Central China after data assimilation. This seems to be insufficient.

How about the inter-annual variability of MODIS AOD over January 2015-2017? Are they consistent with surface PM2.5 measurements? Please clarify.

**Response:**

Actually it's difficult to make judge of the assimilation experiment improvements by using MODIS/AERONET AOD data, as the vertical profiles and the assumptions of optical properties in the model can't be evaluated at this stage. According to the other reviewer's suggestion, we decided to remove the entire session relevant with MODIS/AERONET AOD.

(5) Page 18, Line 23:

In the statement "meteorological conditions might be totally different from 2016 to

2017", "totally" is a very strong word, however, it is not clear how different 2017 meteorological conditions are different from normal wintertime conditions with Siberian Highs. I have the same comment for Page 20, Line 13, Line 19, the word "totally" is not helpful. I suggest use more quantitative statements, for example, higher temperature by how much?

**Response:**

Thanks for the suggestion! Yes, those statements are too strong and not quantitative. We have added the statistic of the meteorology differences by regions in Table 5 and also changed the statements in the texts accordingly.

**Table 5.** Statistics of the meteorological differences by region for January 2015, 2016 and 2017.

| | PBLH (meter) | | | PSFC (Pa) | | | T2 (degree) | | | RH2 (%) | | | WS10 (m/s) | | |
|---|---|---|---|---|---|---|---|---|---|---|---|---|---|---|---|
| | 2016 -2015 | 2017 -2016 | 2017 -2015 | 2016 -2015 | 2017 -2016 | 2017 -2015 | 2016 -2015 | 2017 -2016 | 2017 -2015 | 2016 -2015 | 2017 -2016 | 2017 -2015 | 2016 -2015 | 2017 -2016 | 2017 -2015 |
| NCP | 27.9 | -26.7 | 1.2 | 138.5 | -30.2 | 108.4 | -4.9 | 3.3 | -1.6 | 3.0 | 5.1 | 8.1 | 1.15 | -0.78 | 0.37 |
| NEC | 22.7 | 35.3 | 58.0 | 117.0 | -58.7 | 58.3 | -4.9 | 4.4 | -0.5 | -5.7 | 3.1 | -2.6 | 0.96 | -0.38 | 0.57 |
| EGT | 13.6 | 1.1 | 14.7 | 28.0 | -8.4 | 19.7 | -4.0 | 4.0 | 0.0 | 10.0 | -14.9 | -4.9 | 0.14 | -0.50 | -0.36 |
| XJ | -0.9 | -13.8 | -14.7 | 151.3 | -43.1 | 108.1 | -1.3 | -0.8 | -2.1 | 5.5 | -2.1 | 3.4 | 0.36 | -0.14 | 0.22 |
| FWP | 67.7 | -51.6 | 16.1 | 64.6 | -12.2 | 52.4 | -3.8 | 3.4 | -0.4 | 2.8 | -0.8 | 2.0 | 1.05 | -1.00 | 0.06 |
| SB | 9.8 | -13.2 | -3.4 | -15.9 | 15.9 | 0.1 | -2.4 | 2.5 | 0.2 | 3.9 | -1.8 | 2.0 | 0.43 | -0.02 | 0.41 |
| CC | 34.8 | -56.6 | -21.9 | 82.8 | -53.2 | 29.6 | -2.5 | 2.1 | -0.3 | 10.8 | 0.7 | 11.5 | 0.60 | -0.07 | 0.53 |
| YRD | 64.7 | -22.0 | 42.7 | 77.1 | -27.8 | 49.2 | -1.7 | 1.9 | 0.2 | 7.8 | 2.5 | 10.3 | 0.89 | -0.40 | 0.49 |
| PRD | -36.1 | 8.2 | -27.9 | -16.2 | -60.1 | -76.3 | -0.5 | 2.4 | 1.9 | 11.9 | -8.7 | 3.2 | 0.94 | -0.48 | 0.46 |

(6) Page 20, Line 15:

What does "higher RH (thus more reactions)" mean? How higher RH lead to more chemical reactions? Please clarify.

**Response:**

Yes, in our updated chemistry scheme with newly added heterogeneous reactions ($SO_2$, $NO_2$ and $NO_3$ relevant), higher RH may cause higher uptake coefficients thus more reactions. In the new scheme, the lower and upper limits were used to present a range of uptake coefficient values in the laboratory measurements which were applied when RH is lower than 50% and higher than 90%, respectively. The values in the 50-90% RH range are linearly interpolated based on the two limits. It means when RH exceeds 50%, the uptake coefficients would increase quickly. The details are in Chen (et al. 2016).

In relatively humid regions, such as Central China, Yangtze River Delta and Pearl River Delta, the inter-annual changes of $RH_2$ reached ~10%, which are expecting to affect the heterogeneous reactions.

(7) Captions of Figure 7 and Figure 9: Please indicate here that the comparisons are for the January month.

**Response:**

Clarified in the caption.

(8) Some technical corrections:

Page 2, Line 8 - "modeled PM2.5 are" should be "modeled PM2.5 concentrations are"

Page13, Line 7 - "reflect combining effects" should be "reflect combined effects "

Page 14, Line 11 - "the 2010 EI" should be "the 2010 emissions"

Page 19, Line 17 - "the emission in : : :" should be "the emissions in : : :"

**Response:**

Corrected accordingly.

[revised manuscript text omitted]

---

## Author Comment (AC2) · 19 Feb 2019

Retrospective analysis of 2015-2017 winter-time PM2.5 in China: response to emission regulations and the role of meteorology

Manuscript ID:    acp-2018-890

Authors: Chen, et al.

**Reviewer 2**

The manuscript by Chen et al. evaluates PM2.5 in January in 2015 - 2017 in China.

Data assimilation is used to construct a reanalysis that well matches the available observations of surface-level PM2.5 concentrations. The difference between this and a Non-forced simulation is cleverly used to parse out the separate roles of emissions from meteorology in variability of PM2.5 concentrations, which are significant across these 3 years (although they authors refer to these as trends – which I think is better described as inter annual variability). This is important and valuable work, because it helps identify reasons why emissions control strategies may or may not have an immediately visible impact. As such the topic and scope are suitable for ACP. The manuscript includes some additional analysis of AOD, but it is somewhat secondhand, not as well supported in terms of model accuracy, and doesn't particularly add to the focus of the paper, which I suggest remain on surface-level PM2.5 concentrations. This would provide space for the authors to provide more details on other aspects of their modeling and assimilation methods, which in several places are too abbreviated or presented without sufficient background or justification. The major scientific weakness in this work is likely the use of out-dated emissions (from 2010), given the rate at which species like SO2 and NOx are known to be changing since 2010 in China. Below I provide more detailed comments on these and other aspects to address prior to publication in ACP.

**Response:**

Thanks for the valuable and insightful suggestions! We have made several changes accordingly.

1. Removed the contents relevant with AOD comparisons.

2. Added more details on modeling and assimilation methods.

3. Rerun the assimilation experiment with looser filter criteria, in which the interannual changes were better captured.

4. Added the introduction of model deficiencies, including the imperfection of chemistry mechanism, and also the inaccuracy of 2010-MEIC EI for the study period. The impacts of the deficiencies on analysis are also discussed.

Please see the itemized responses as below. Revised manuscript is after the response letter.

Comments: The grammar needs work throughout. This might mean adding another co-author or hiring an editing service. The number of corrections needed (nearly every sentence) are far too extensive for me to detail here.

The abstract would be improved by considering some of the following suggestions:

- lead with a broader, introductory statement

- avoid jargon when possible

- try to provide a mix of qualitative and quantitative results. Presently, only qualitative results are reported

- end with a statement about the bigger impacts of this work

- reduce the overall length – currently it goes too much into details of the methods, without quantitatively summarizing the most important conclusions.

**Response:**

We have hired the editing service from American Journal Experts (www.aje.com) and edited the manuscript thoroughly. We also revised the abstract accordingly.

**Q1.** 4.22: I believe that this statement regarding the inorganic PM2.5 in this region has been known for some time.

**Response:**

Yes, thanks for pointing out this. We have added relevant references here and removed the repeated sentences.

**Q2.** 7.1-3: There is considerable debate in the literature regarding the reactions that may be leading to high concentrations in haze events. The ones included here are based on assumptions of pH that may not be correct. Other recent works have e.g. suggested

HCHO may play a role. In short, I recommend the authors review the relative literature on this topic (which should be easy to find, several very high-profile papers). Even if they decide to stick with their current

mechanism (which I think is acceptable, given their is yet to be scientific consensus on this issue), discussion is warranted, both in the introduction and consideration of sources of errors towards the end of the paper.

**Response:**

We have added relevant references of the recently proposed chemistry mechanisms and also discussed the potential impacts on the uncertainties of analysis.

"In addition to the uncertainties in emission inventories, deficiencies in the model chemistry can also cause model uncertainties. Increasing numbers of observations have revealed that anthropogenic emission-relevant aerosol species, such as sulfate, nitrate and ammonium (denoted as SNA), are the predominant inorganic species in the wintertime $PM_{2.5}$ in China (Wang *et al.*, 2014c; Yang *et al.*, 2015). Various reaction paths during haze events have also been proposed (e.g. Zheng *et al.*, 2015; Cheng *et al.*, 2016; Wang *et al.*, 2016; Li *et al.*, 2017; Moch *et al.*, 2018; Wang *et al.*, 2018; Shao *et al.*, 2019). For example, Moch *et al.* (2018) used a 1-D model and revealed the importance of aqueous-phase chemistry of HCHO and S(IV) in cloud droplets by forming a S(IV)-HCHO adduct, hydroxymethane sulfonate. Shao *et al.* (2019) implemented four heterogeneous sulfate formation mechanisms (via $H_2O_2$, $O_3$, $NO_2$, and transition metal ions on aerosols) into GEOS-Chem model which partially reduced the modeled low bias in sulfate concentrations. However, a scientific consensus regarding the importance of the reaction paths has not yet been reached partially due to the uncertainties of aerosol liquid water content, pH, and ionic strength etc. The original WRF/Chem model with either the Goddard Chemistry Aerosol Radiation and Transport (GOGART, Chin *et al.*, 2000, 2002) or the Model for Simulating Aerosol Interactions and Chemistry (MOSAIC)-4BIN aerosol scheme failed to reproduce the highest $PM_{2.5}$ concentrations; it is assumed that this failure is due to missing heterogeneous/aqueous reactions. In Chen *et al.* (2016, hereafter Chen16), we added three heterogeneous reactions ($SO_2$-to-$H_2SO_4$ and $NO_2$/$NO_3$-to-$HNO_3$) to the WRF/Chem model based on the MOSAIC-4BIN aerosol scheme. Although the reaction paths may still not be comprehensively understood, the new MOSAIC-4BIN aerosol scheme significantly improved the simulation of sulfate, nitrate, and ammonium on polluted days in terms of the concentrations of those species and their partitioning."

"It is worth noting that there are uncertainties in the simulation/assimilation processes. There are three sources of uncertainties in the NO_DA simulation. First, the emission inventories in the NO_DA simulations are obviously not accurate, which may introduce uncertainties into the analysis……Second, the meteorological IC/BC conditions in the NO_DA simulations, which were obtained from GFS 6-hr analysis data, also have uncertainties……Third, the deficiencies associated with the chemistry in the model also generate uncertainties, including missing reactions and the inaccurate parameterization of reactions. These three aspects all originate from the imperfections of current forward models."

**Q3.** Figure 2: It's not clear what this shows. What are "PM2.5 emissions"? Since many of the species contributing to PM2.5 are gas-phase precursors, which don't necessarily completely transform into aerosol, I'm quite puzzled. Perhaps it is a plot of a subset of

PM2.5 precursor emission species, but that isn't made clear.

**Response:**

The original figure shows the primary $PM_{2.5}$ emission, corresponding to the total of BC, OC, sulfate, nitrate and other unspecified $PM_{2.5}$ emissions. We have clarified in the figure caption. To be more comprehensive, the emission spatial distributions of $PM_{2.5}$ precursors, including $SO_2$, $NO_x$ and $NH_3$ have also been added in Figure 2.

**Q4.** 8.1 and other locations (e.g., 14.11): The emissions used in this work are quite outdated. It is documented in several studies of emissions in China that SO2 emissions have been decreasing since around 2009, and NOx emissions since around 2011 or 2012. These previous studies need to be cited, and considered in the present work. It is certainly a significant source of error worth considering.

**Response:**

The 2010-MEIC EI is the only emission inventory that publicly available when the study was conducted. We have added the relevant references on the emission reductions of $SO_2$ and $NO_x$ in the text.

"As the Chinese government has implemented strict control strategies to ensure an improved air quality during the winter season since 2013, significant reductions in emissions, including primary PM and precursor compounds ($SO_2$ and $NO_x$), in regions with the strict implementation of these policies relative to the year 2010 are expected for our simulation period. A reduction in $SO_2$ pollution of approximately 50% was observed from 2012-2015 for the North China Plain from OMI satellite data (Krotkov *et al.*, 2016). National anthropogenic emission reductions of approximately 67%, 17%, and 35% from 2012-2017 for $SO_2$, $NO_x$, and $PM_{2.5}$, respectively, were assumed by the bottom-up EI methodology (Zheng *et al.*, 2018). However, the expansion and relocation of the energy industry caused emission increases in northwestern China (Ling *et al.,* 2017). In addition, the uncertainties of allocated emissions in the winter season will be much larger than those in other seasons. For example, Zhi *et al.* (2017) conducted a village energy survey and revealed an enormous discrepancy in the amount of rural raw coal used for winter heating in northern China, implying an extreme underestimation of rural household coal consumption by the China Energy Statistical Yearbooks. These changes and uncertainties of emissions in the model would introduce errors into the NO_DA simulation. However, the inhomogeneous spatial changes and large uncertainties of seasonal allocations made it difficult to simply scale the original emission inventory for our study period."

"In the NO_DA case, the model results are overpredicted in SB, NCP and CC for all three months, while the overestimations are more severe in SB. The NO_DA case generally overestimates (underestimates) the surface $PM_{2.5}$ in NCP, SB and CC (XJ and FWP) in the three years, potentially indicating that the 2010 emissions are not appropriate for the 2015-2017 simulations with overestimations (underestimations). As discussed in section 2.1, the large area of overestimation is consistent with the national reductions in $SO_2$, $NO_x$

and PM$_{2.5}$ anthropogenic emissions (Zheng *et al.*, 2018); however, the underestimations in XJ and FWP also indicate the introduction of new emission sources to these two regions."

"It is worth noting that there are uncertainties in the simulation/assimilation processes. There are three sources of uncertainties in the NO_DA simulation. First, the emission inventories in the NO_DA simulations are obviously not accurate, which may introduce uncertainties into the analysis. Although the basic assumption required only that the emissions stay the same throughout the three years, emission inventory uncertainty-induced errors would be offset in the subtraction process when calculating the year-to-year differences. However it did generate uncertainties. For example, the emissions in SB, CC and PRD were generally overestimated (Fig. 3), which means that the variations in the ambient concentration might have been artificially amplified considering the meteorology impacts (Fig. 6c). In contrast, the emissions in XJ and FWP were underestimated (Fig. 3), and thus, the changes in the ambient concentrations due to meteorological conditions in these two regions might have diminished. From this point of view, if the fixed emissions are more accurate in those years, the results would be more reliable. In the case where "real" emissions are not available and the purpose is to evaluate the contribution of those emissions, uncertainties will be unavoidable and should be emphasized carefully."

**Q5.** 9.2: Not sure why capital pi notation is used here, as that means product, where the definition is in terms of a sum.

**Response:**

Changed to $M_{PM_{2.5}}$ instead.

**Q6.** 9.10-12: I don't understand how this works – can it be explained further? The way in which measurements of total PM2.5 are used to adjust concentrations of specific species should be clarified, even though it comes from an earlier work, if only briefly.

**Response:**

We have added the following explanation in the text (highlighted in blue).

"Since only surface PM$_{2.5}$ total mass observations were assimilated to analyze the 3D mass mixing ratios of 24 aerosol variables, the 3DVAR problem was underconstrained. Due to the lack of species and vertical information provided by the observations, the only mathematical solution is to utilize prior information from the model background. In the GSI system, the distribution of the analysis increments (the difference between the analysis and background) onto the different species was mostly model driven with the observation and background error covariance matrices acting as the main constraints."

"Similar to Jiang13, the background error covariance (BEC) statistics for each analysis variable required by the 3DVAR algorithm were computed by utilizing the NMC method (Parrish and Derber, 1992) based

upon the one-month WRF/Chem forecast for January 2015. No cross-correlation between different species was considered. The standard deviations and horizontal/vertical correlation length scales of the background errors (separated for each aerosol species) were calculated using the method described by Wu *et al.* (2002). These data were used as constraints for the distributions of the PM components."

**Q7.** 9.20: What is the origin of this assumption regarding error? The relative error component of 0.75% seems very small. Other parameters such as gamma and L, the maximum concentration threshold (500) or analysis increment (120) are similarly introduced without much explanation. I recognize these values have been used before, but still more explanation here would be appreciated.

**Response:**

We have added the original reference in the text. Regarding the parameter of error calculation, we have added the explanations for those parameters as below.

"The observation error covariance matrix **R** in Eq. (1) contains both measurement and representativeness errors. Pagowski *et al.* (2010) used a measurement error ($\varepsilon_0$) of 2 µg m$^{-3}$. To associate higher PM$_{2.5}$ values with larger measurement errors, S12 defined the measurement error as $\varepsilon_0 = 1.5 + 0.0075 \times M_{PM_{2.5}}$, where $M_{PM_{2.5}}$ denotes an AIRNow PM$_{2.5}$ observation and the units of each term are µg m$^{-3}$. According to the PM$_{2.5}$ Auto-Monitoring Instrument Technical Standard and Requirement (China National Environmental Monitoring Center, 2013), three continuous online monitoring methods, namely, a beta-ray plus dynamic heating system, a beta-ray plus dynamic heating system plus light scattering system, and a tapered element oscillating microbalance plus filter dynamic measurement system, are used at the national monitoring sites to satisfy the requirements that the display resolution should be less than 1 µg m$^{-3}$ and the error should be less than 5 µg m$^{-3}$(within 24 hours). To reflect the confidence in the hourly observations, the measurement error $\varepsilon_0$ in this study is defined as $\varepsilon_0 = 1.0 + 0.0075 \times M_{PM_{2.5}}$, where $M_{PM_{2.5}}$ denotes a PM$_{2.5}$ observational value (unit: µg m$^{-3}$).

"Representativeness errors reflect the inaccuracies in the forward operator and in the interpolation from the model grid to the observation location. Elbern *et al.* (2007), Pagowski *et al.* (2010), S12 and Jiang13 defined the representativeness error ($\varepsilon_r$) as

$$\varepsilon_r = \gamma \varepsilon_0 \sqrt{\frac{\Delta x}{L}}, \qquad (3)$$

"where $\gamma$ is an adjustable parameter scaling $\varepsilon_0$ ($\gamma = 0.5$ was used here), $\Delta x$ is the grid spacing (40 km in our case) and L is the radius of influence of an observation (set to 2 km for urban sites). These parameter settings were based on the performance of sensitivity tests."

The criteria for filter process are from two aspects, including the stability of DA optimization step and the computing efficiency. The original criteria were mostly set for operational runs. For research purpose, we

have made tests of different filter process and found that looser filter can really improve the assimilation results. Here in the revised manuscript, only PM$_{2.5}$ observations larger than 1000 μg m$^{-3}$ (the maximum display limit of the monitoring system) were deemed unrealistic in the filter process and observations leading to deviations exceeding 500 μg m$^{-3}$ were omitted in the assimilation process for the stability of the assimilation optimization. Besides, in the original assimilation experiment observational sites located in grids with elevation greater than 500 meter (Above Sea-Level) were not used; to better utilize those data, we chose to interpolate them to the lowest model level for assimilation.

Below shows the observed, original and updated assimilated monthly average of PM$_{2.5}$ concentrations (unit: μg m$^{-3}$) for January in 2015 (Left), 2016 (middle) and 2017 (right). The most prominent improvements are shown for the hotspots in Xinjiang (Region d) and Fenwei Plain (added as Region e).

[Figure]

**Figure S1.** Observed and modeled monthly average of PM$_{2.5}$ concentrations (unit: µg m$^{-3}$) for January in 2015 (Left), 2016 (middle) and 2017 (right). (a) Observation, (b) Original CONC_DA, (c) New CONC_DA with looser filter. Regions defined in red rectangles are: a-NCP (North China Plain), b-NEC (Northeastern China), c- EGT (Energy Golden Triangle), d-XJ (Xinjiang), e-Fenwei Plain (FWP), f-SB (Sichuan Basin), g-CC (Central China), h-YRD (Yangtze River Delta), i-PRD (Pearl River Delta).

In the updated assimilation experiment, the interannual changes are also better captured as shown below. Those improvements make our analysis more solid, especially for the Xinjiang region and Fenwei Plain. The updated figures and discussions are highlighted in blue in the revised manuscripts.

[Figure]

**Figure S2.** Observed and modeled PM$_{2.5}$ ambient concentration changes for 2016-2015 (left), 2017-2016 (middle) and 2017-2015 (right). (a) Observation, (b) Original CONC_DA, (c) New CONC_DA with looser filter. (Unit: $\mu g\ m^{-3}$)

**Q8.** 10.12: Omission of cross-correlation between species like ammonium and nitrate seems critical – how does this impact the results?

**Response:**

Actually, it would be critical if the individual species observations (such as ammonium, sulfate, and nitrate) were assimilated while the correlation between them were not considered. However, due to the lack of species observation available, the adjustments of species were entirely from the prior information form the model background. As the species partitioning were reasonably considered in the updated chemistry scheme, it's expected that the correlation among the species are well presented and the omission of cross-correlation would bring very small impacts to the assimilation results. In the future studies, with more species observations, the comparisons of individual species should be addressed.

**Q9.** Section 2.5: The method used for estimating the impact of meteorology separately from emissions seems sound; I'm not sure the extended explanation (by way of comparison to radiative forcing calculations, etc.) is needed and suggest simply removing the first paragraph of Section 2.5 and jumping directly into the statement of how this study was carried out.

**Response:**

Thanks for the suggestion. We have removed most contents of the first paragraph of section 2.5.

**Q10.** Fig 4: The model performance seems good after the assimilation. One small question though – it seems like the residual bias in the CONC_DA case is most often negative.

Is there a reason for this? I would have expected that, given the initial simulation is biased high, the analysis would not necessarily be unbiased, but would similarly have a small slight high bias (owing to the constraint term in the cost function).

**Response:**

Thanks for pointing out this! Actually I'm thinking that's due to the data filter process in which high values of observations were not assimilated; but all the data with observations available (including high values) are compared and shown in the figure.

As answered in Q6, we have chosen looser filter criteria that only PM$_{2.5}$ observations larger than 1000 µg m$^{-3}$ were deemed unrealistic and observations leading to deviations exceeding 500 µg m$^{-3}$ were omitted. Below the updated figure shows that the analysis is slightly high bias as expected.

[Figure]

**Figure S3.** Time series of the statistics between the model simulations and observations. Red lines-CONC_DA minus observations, blue lines –NO_DA minus observations. Statistics include the number of data pairs for comparison, the MEAN-mean bias, the STDV- standard deviation, and the RMS-root mean square error. Left-2015, middle-2016, right-2017. (units are µg m$^{-3}$ for MEAN, STDV and RMS).

**Q11.** Fig 5: Figs 5b and 5c could be omitted or moved to supporting information.

**Response:**

Thanks for the suggestion. Figs 5b and 5c were moved to the supporting information.

**Q12.** 15.24: Why not use level 2 data?

16.7: I didn't follow nor agree with the logic behind this statement. Why not include statistics of the AOD comparison in a table, or directly on the figures themselves (there is room in the white space). Also, it seems that DA only serve to decrease the AOD, not increase it, even in areas.

16.14: Let's be honest – the DA didn't "didn't correct the bias" – it made the bias even worse, for 6 out of the 9 sites. This won't go un-noticed by the readers, so it might was well be presented fairly.

Overall, the AOD analysis was on the weaker side, the connections to the policy and haze questions not as clear, and the model performance not as good (especially for

AERONET). I would suggest the authors consider dropping AOD related content entirely, unless some more satisfactory explanations can be included.

**Response:**

Thanks for the suggestion! Actually it's difficult to make judge of the assimilation experiment improvements by using MODIS/AERONET AOD data, as the vertical profiles and the assumptions of optical properties in the model can't be evaluated at this stage. According to your suggestion, we decided to remove the entire session relevant with MODIS/AERONET AOD.

**Q13.** Section 4: Evaluation of concentrations in January alone of 3 consecutive years is not sufficient enough time range for a "trends" analysis. We do get some sense of interannual variability though, which is interesting. It is just mislabeled. For example, every place that says something like "decreasing trend from 2015 to 2016" should say "decrease from 2015 to 2016", as the long-term trend hasn't been determined. If the authors really do wish to study trends, they should have considered years such as 2005, 2010 and 2015. Or if they can't go back that far, owing to data availability,

2012, 2014, and 2017. That would start to be close to enough years to make a trend analysis. My overall suggestion would actually be to remove section 4.1 entirely, at least the second paragraph on AOD. I also wonder if any of these years happened to be impacted by PM2.5 transport more than others, e.g. from fires?

**Response:**

Thanks for the suggestion! The second paragraph on AOD was removed. We have changed the word "trends" to "interannual changes".

Actually Chinese government has implemented strict control strategies on crop/biomass burning, thus the local and central monitoring teams were mandatory on site to prevent fire events. Besides, no big natural forest fires occurred during our study period. For these reasons, we assumed that the interannual changes of $PM_{2.5}$ are mostly driven by meteorology and anthropogenic emissions.

**Q14.** 18.13: I don't understand what is meant by the sentence beginning "Thus only. . ."

**Response:**

Revised the sentence to "The ambient response from 2015-2017 is contradictory if considering only the reductions in emissions and omitting the changes in meteorological conditions."

**Q15.** Fig 8: Suggest removing this figure; it is barely discussed, and doesn't add much. The analysis of PM2.5 surface concentrations is sufficient and also more convincing, since the model performance is better.

**Response:**

Thanks! Figure 8 had been removed.

**Q16.** 20.1: I think this is an important point – if the authors are using this approach to separate emissions impacts from meteorology in the observed dataset, then it is critical the relative changes in the total assimilation experiment (row b) match the observations.

**Response:**

Thanks for pointing out this! We have rerun the assimilation experiment and the updated results did reproduce the interannual changes (as shown below).

[Figure]

**(c)  New CON_DA with looser filter**

[Figure]

**Figure S2.** Observed and modeled PM$_{2.5}$ ambient concentration changes for January 2016-2015 (left), 2017-2016 (middle) and 2017-2015 (right). (Unit: µg m$^{-3}$).

**Q17.** 21.14: Change "verified" to "evaluated", here and throughout, since in the strict sense of the word, the model has certainly not been verified.

**Response:**

Corrected in the text.

**Q18.** 22.2: I note the authors stop short of including anything regarding the AERONET evaluation in the conclusions – an indication that this part of the manuscript could be removed without impact.

**Response:**

Yes, the contents relevant with MODIS AOD and AERONET AOD have been removed.

[revised manuscript text omitted]

---

## Author Response (AR2)

Dear ACP Editor:

We have made all the technical corrections raised by the reviewer, and incorporated them in the revised manuscript.

Thank you very much for your consideration.

Sincerely,
Dan Chen, et al.